# Changes in orogenic style and surface environment recorded in Paleoproterozoic foreland successions

Bo Huang [1] ✉, Man Liu[1], Timothy M. Kusky [1], Tim E. Johnson [1,2], Simon A. Wilde [2], Dong Fu [1], Hao Deng[1] & Qunye Qian[1]

The Earth's interior and surficial systems underwent dramatic changes during the Paleoproterozoic, but the interaction between them remains poorly understood. Rocks deposited in orogenic foreland basins retain a record of the near surface to deep crustal processes that operate during subduction to collision and provide information on the interaction between plate tectonics and surface responses through time. Here, we document the depositional-to-deformational life cycle of a Paleoproterozoic foreland succession from the North China Craton. The succession was deposited in a foreland basin following ca. 2.50–2.47 Ga Altaid-style arc–microcontinent collision, and then converted to a fold-and-thrust belt at ca. 2.0–1.8 Ga due to Himalayan-style continent–continent collision. These two periods correspond to the assembly of supercratons in the late Archean and of the Paleoproterozoic super-continent Columbia, respectively, which suggests that similar basins may have been common at the periphery of other cratons. The multiple stages of orogenesis and accompanying tectonic denudation and silicate weathering, as recorded by orogenic foreland basins, likely contributed to substantial changes in the hydrosphere, atmosphere, and biosphere known to have occurred during the Paleoproterozoic.

Earth is the only known planet that has emergent felsic continents and currently operates in a plate tectonic (mobile lid) mode. It is also the only place known to be (or have been) inhabited, where its habitability is sustained through interactions between the lithosphere, hydrosphere, atmosphere, and biosphere[1–4]. The late Archaean to early Paleoproterozoic was one of the most transformative periods in Earth's history[5–7] (Fig. 1), witnessing the widespread emergence of continents above sea level, global cratonisation, the onset of the supercontinent cycle and, perhaps, the establishment of global plate tectonics[8–10]. In terms of near-surface processes, this period is associated with the formation of large-scale banded- and granular-iron formations (BIF and GIF) and graphite–phosphorite mineralisation, the Huronian glaciation, the Great Oxidation Event (GOE), and the emergence of multicellular life (eukaryotes)[11–14] (Fig. 1c). Although there is a growing consensus that (global) plate tectonics was operative in the late Archaean to early Paleoproterozoic[10,15–17], whether and how the orogenic style might have differed from modern subduction-to-collision systems and the feedback mechanisms between solid and surficial reservoirs are not well understood.

The erosion, transportation and deposition of clastic sediments during tectonic uplift and weathering of continental rocks likely mediated the long-term carbon cycle in deep time[18–20]. Foreland basins form between fold-and-thrust belts and the craton interior in typical orogens, representing an important sedimentary archive of the processes associated with continent–continent or arc–continent collision at convergent plate boundaries[21]. These successions characterise the

[1]Badong National Observation and Research Station of Geohazards, State Key Laboratory of Geological Processes and Mineral Resources, Center for Global Tectonics, School of Earth Sciences, China University of Geosciences, Wuhan 430074, China. [2]School of Earth and Planetary Sciences, the Institute for Geoscience Research, Timescales of Mineral Systems Group, Curtin University, Perth, WA 6102, Australia. ✉e-mail: bohuang@cug.edu.cn

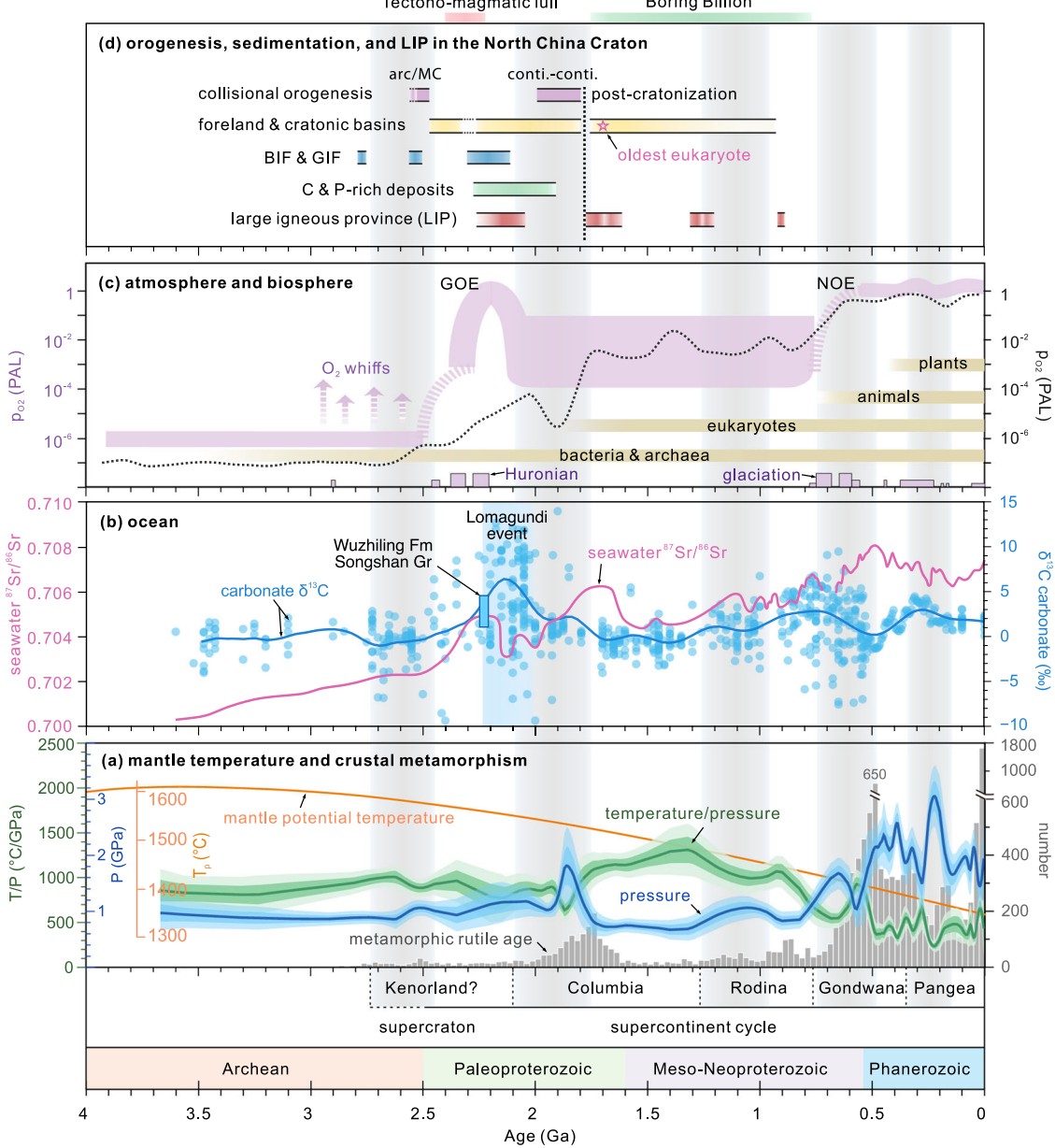

**Fig. 1 | Some key geological proxies signifying the secular change in Earth's solid and surface systems. a** Mantle temperature[90], metamorphic rutile[10], and 10% LOESS bootstrapping smoothing curves of thermobaric ratios (*T/P*) and peak pressure (P) of metamorphic rocks (updated from datasets in refs. 8, 88). **b** Normalised seawater [87]Sr/[86]Sr[77] and marine carbonate δ[13]C[91]. Note the significant increase of seawater Sr at 2.5–2.3 Ga and 2.0–1.8 Ga, and the large positive excursion of carbonate δ[13]C (Lomagundi event). **c** Estimated atmospheric oxygenation[7,92] and biosphere evolution[12]. The dotted dark curve of atmospheric oxygenation is based on machine learning[92]. GOE Great Oxidation Event, NOE Neoproterozoic Oxidation Event. PAL present atmospheric level. **d** Late Neoarchean–early Paleoproterozoic and mid-Paleoproterozoic two-generation orogenesis, diagnostic sediments, and LIPs in the North China Craton, MC microcontinent.

source-to-sink history of detritus produced during the uplift and unroofing of orogenic crust[22], and thus have the potential to chart any changes in orogenic style and attendant near-surface processes.

Here, we document a well-preserved early Paleoproterozoic foreland succession (the Songshan Group) in the North China Craton. The succession was deposited in a foreland basin related to a ca. 2.5–2.47 Ga Altaid-style arc–microcontinent collision, and later converted to a fold-and-thrust belt during ca. 2.0–1.8 Ga Himalayan-style collisional orogenesis. Integrating geochronology and structural datasets, we provide constraints on the distinct style of orogenesis and plate tectonics that characterised the Neoarchean and Paleoproterozoic. We then link the processes of mountain building, crustal uplift, denudation-sedimentation, and silicate weathering to dramatic changes in the hydrosphere, atmosphere, and biosphere during the Paleoproterozoic.

## Results and discussion
### Geological background
The North China Craton has been subdivided into the Eastern and Western blocks, separated and margined by several Neoarchean to Paleoproterozoic orogenic belts that record its incorporation into the Columbia supercontinent (Fig. 2a, b)[23–26]. The major north–south trending orogenic belt in the central North China Craton, variably termed the Neoarchean Central Orogenic Belt[24] or the Paleoproterozoic Trans-North China Orogen[27], marks the assembly of the Western and Eastern blocks and intervening arcs (Supplementary Note).

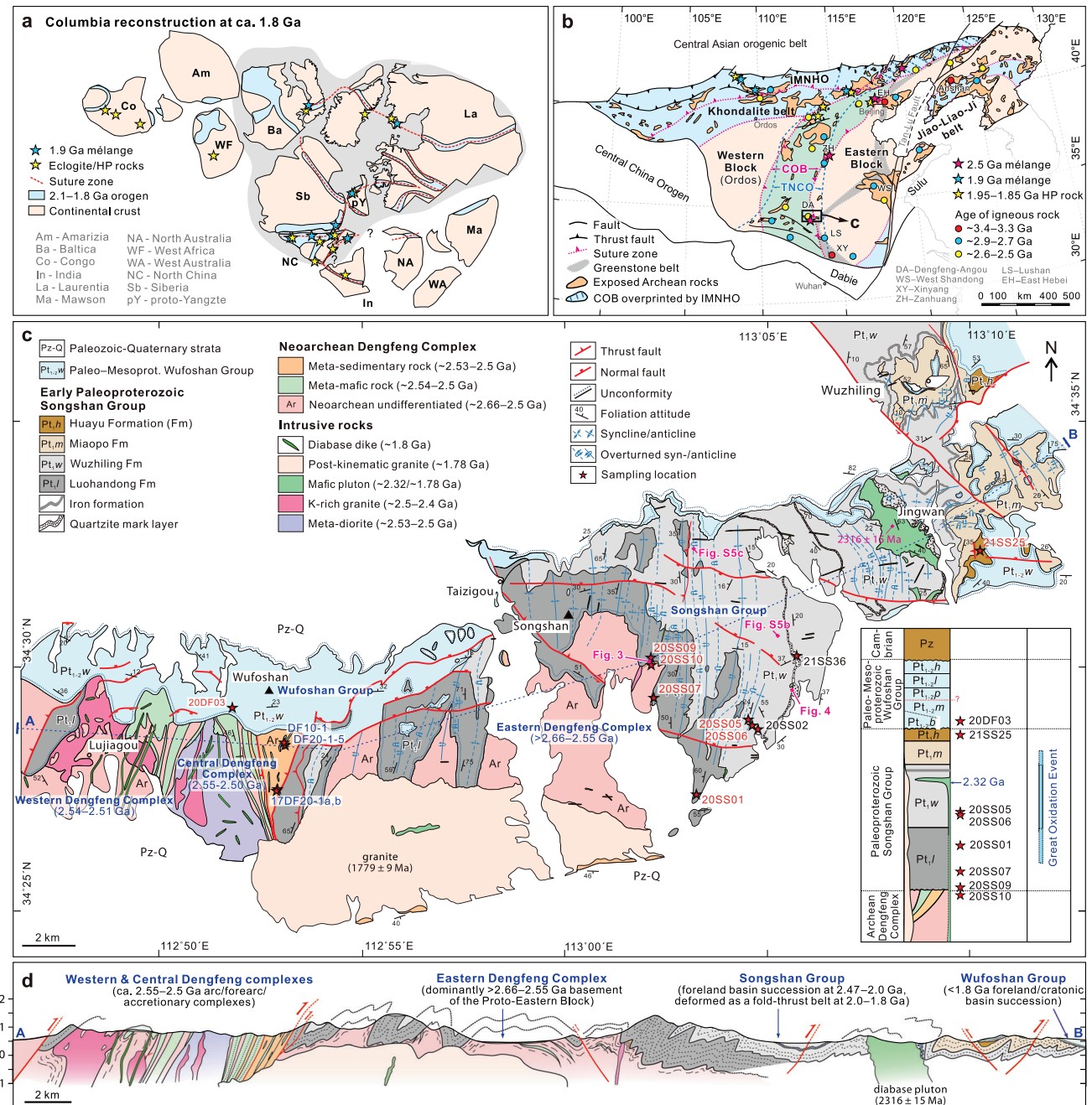

**Fig. 2 | Geological map and cross-section. a** Reconstruction of the Columbia supercontinent (modified from ref. 72). **b** Tectonic framework of the North China Craton (modifed from ref. 25). Major magmatic and metamorphic events are labelled[25,32,45]. IMNHO, Inner Mongolia-North Hebei orogen. COB, Central Orogenic Belt. TNCO, Trans-North China Orogen. **c** Geological map and simplified stratigraphic column of the Dengfeng region showing litho-structural units and strata (modifed from ref. 33). **d** Cross-section showing the structural relations of different litho-structural units. HP high pressure.

In the southern segment of the central orogen, the Songshan Group is a well-exposed Paleoproterozoic sedimentary succession considered to have formed within a foreland basin[24,28], although details regarding its provenance and time of deposition and deformation remain incompletely constrained. It unconformably overlies Neoarchean TTG gneisses and metavolcano-sedimentary (greenstone) assemblages of the Dengfeng (-Angou) Complex, and is unconformably overlain by the unmetamorphosed Paleo- to Meso-proterozoic sedimentary succession of the lower Wufoshan Group (Fig. 2c)[28–30]. The Dengfeng Complex is interpreted to comprise a subduction-related arc/forearc unit in the west, and an ocean plate stratigraphy-dominated accretionary unit in the central part[16,29,31],

which together are interpreted to represent a Neoarchean paired metamorphic belt, and to record seafloor spreading, oceanic subduction, to an arc–microcontinent collision event[32]. A metasedimentary and gneissic unit in the eastern Dengfeng Complex is regarded as a passive margin sequence with cratonic basement[16,29]. The greenstone assemblage of the central Dengfeng Complex is strongly deformed, with minor small-scale tight-isoclinal folds formed during top-to-the-NE/E thrusting[29]. Available U–Pb isotopic dating of metamorphic zircon and titanite has identified a ca. 2.5 Ga episode of amphibolite facies metamorphism[31,32], although the influence of the Paleoproterozoic events on the rocks is poorly constrained.

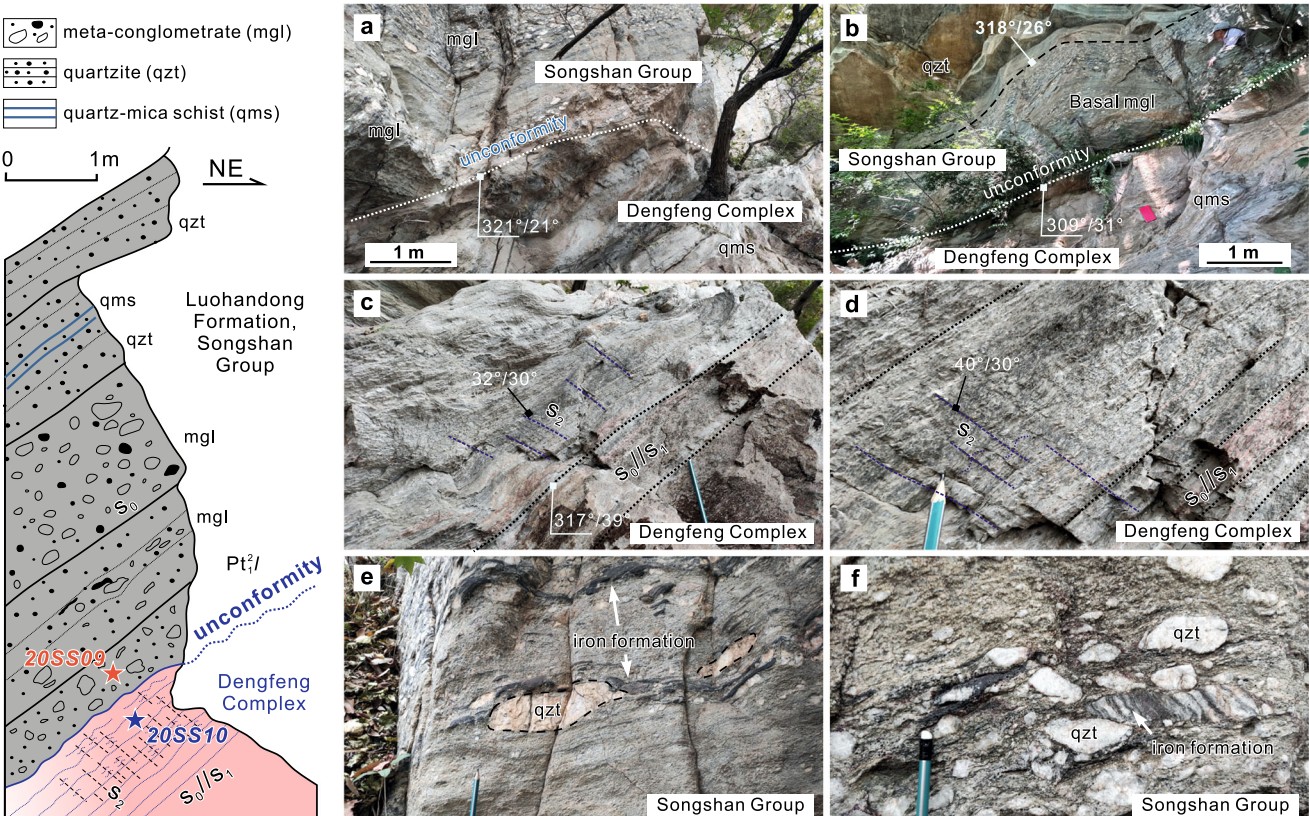

**Fig. 3 | Unconformity between the Paleoproterozoic Songshan Group and the underlying Neoarchean Dengfeng Complex. a, b** Field photographs showing the unconformity. **c, d** Quartz mica schists of the Dengfeng Complex near the unconformity. **e, f** Deformed and elongated clasts in the basal meta-conglomerate of the Songshan Group. The left schematic lithological column is modified from ref. 33.

The Songshan Group mainly comprises metamorphosed submarine siliciclastic and chemical sedimentary sequences[33,34]. The group has been regarded as a ca. 2.5–2.3 Ga peripheral foreland basin sequence unconformably overlying both the Central Orogenic Belt and the Eastern Block, which was formed by the collision between an arc that lay to the west of the Eastern Block[24,25,29]. In this model, the foreland basin probably evolved to become a retro-arc foreland or rift basin in the middle-late Paleoproterozoic, during which time the upper part of the group was deposited[25,29]. An alternative model considers the entire Songshan Group to represent a significantly younger (maximum depositional ages of 2.35 Ga in the lower section and 1.96–1.78 Ga in the upper section) retro-arc foreland basin that developed behind an Andean-type continental arc along the western margin of the Eastern Block, then was incorporated into the Paleoproterozoic (ca. 1.97–1.85 Ga) Trans-North China orogen during collision between the Western and Eastern blocks of the North China Craton[27,28].

We undertook detailed structural analysis and collected samples from the Songshan Group, the underlying Dengfeng Complex, and the overlying Wufoshan Group, for petrological and multi-mineral geochronological analysis (zircon/rutile U–Pb and amphibole/white mica $^{40}Ar/^{39}Ar$ dating). These new data constrain the depositional-to-deformational evolution of the Songshan foreland basin. Furthermore, analysis of detrital and/or metamorphic zircon and rutile identifies distinct metamorphic patterns within the detrital source regions, thereby providing additional insight into the styles of orogenesis and plate tectonics operating during the Neoarchean to Paleoproterozoic.

## Field characteristics and petrology
The Songshan Group is a N–S-striking sequence of metasedimentary rocks that is up to ~1.7 km thick and comprises basal conglomerate,

quartzite, quartz-mica schist, with minor layers of marble and GIF (Fig. 2c, d). The strata are strongly deformed, underwent upper greenschist (locally up to amphibolite) facies metamorphism and were intruded by mafic and felsic dikes/plutons and other igneous bodies[33]. The Songshan Group has been subdivided into four lithological packages termed, from stratigraphic base to top, the Luohandong, Wuzhiling, Miaopo, and Huayu formations[33] (Fig. 2c; Supplementary Fig. 1).

The Luohandong Formation is dominated by a thick (100s of metres) sequence of quartzites (Supplementary Fig. 2a, b), with a 2–5-m thick basal conglomerate that overlies the Neoarchean quartz-mica schists and TTG gneisses of the Dengfeng Complex (Fig. 3). The clasts in the deformed conglomerate are preferentially aligned with the tectonic foliation and consist mainly of quartzite (some bearing magnetite), and quartz-mica schist (Fig. 3e, f). The quartzite is composed mainly of recrystallised quartz with minor muscovite (Supplementary Fig. 3c) and preserves ripple marks, flute casts and cross-bedding (Supplementary Fig. 2b), consistent with a nearshore littoral facies depositional environment.

The overlying Wuzhiling Formation is up to ~750 m thick and composed of quartz-mica schist and quartzite (Supplementary Fig. 2c) with minor marble and GIF interlayers (Supplementary Fig. 1). The quartz-mica schist consists of white mica, quartz, feldspar (mainly K-feldspar and albite), with minor biotite and rutile (Supplementary Fig. 3f, g). The protoliths are dominantly fine-grained siltstone, shale, quartz sandstone, and chemical sediments, which were likely deposited in littoral to neritic environments. The Wuzhiling Formation is intruded by a large diabase pluton with a zircon U−Pb age of ca. 2.32–2.30 Ga[35,36] (Fig. 2c, d; Supplementary Fig. 4), which provides a minimum age for deposition of the lower Songshan Group strata. The

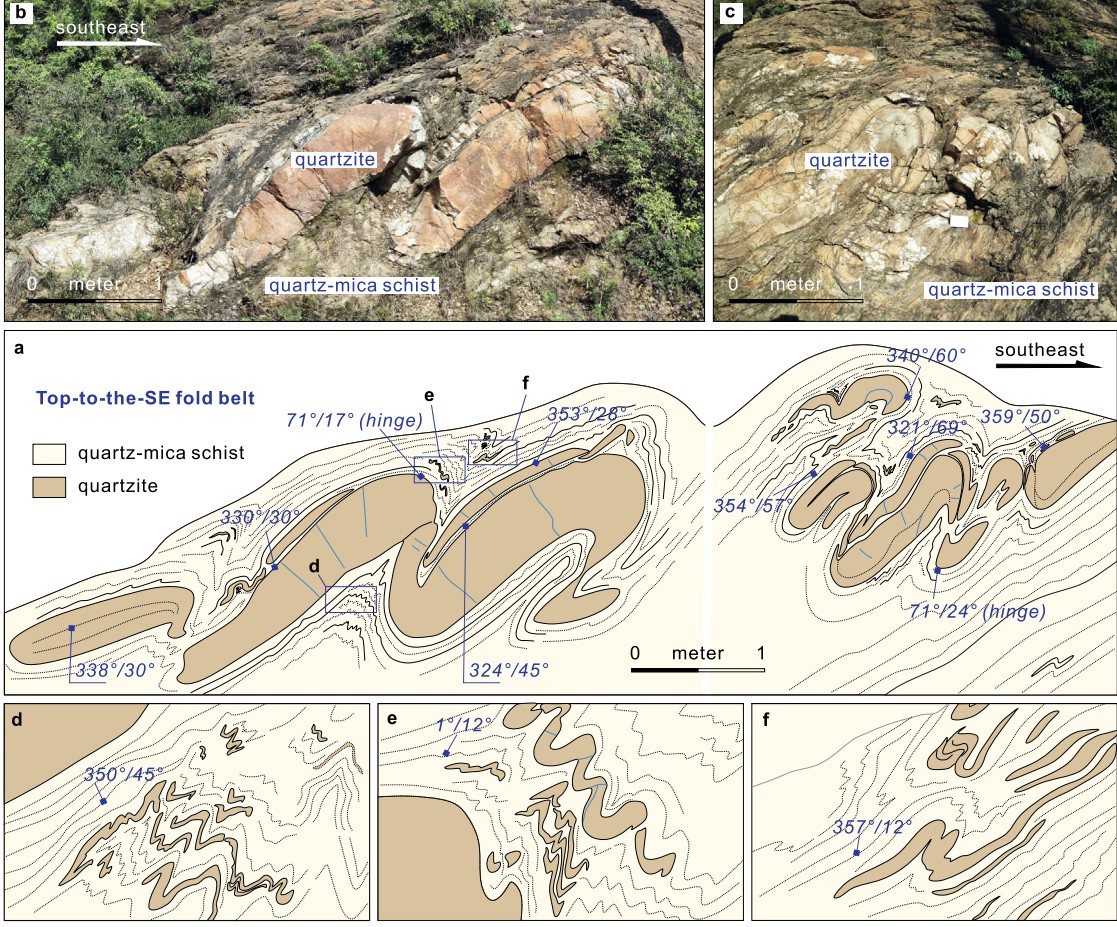

**Fig. 4 | A representative outcrop showing fold structures in the Songshan Group.** Litho-structural mapping (**a**, **d**–**f**) and field photographs (**b**, **c**) showing a representative outcrop of the Songshan fold belt. The folds are overturned or recumbent in which overturned limbs of quartzite layers were thinned and disaggregated due to strong subhorizontal contraction and progressive shearing, with a top-to-the-SE shear and thrust sense. The variation in deformation between quartzite and quartz-mica schist reveals their different competencies. The attitudes (in blue) of foliation and axial planes and hinges of folds are marked.

Miaopo and Huayu formations that comprise the upper Songshan Group are dominated by white to purple banded quartzite and interbedded quartz-mica schist (Supplementary Fig. 2d), with minor layers of marble and phosphorite in the upper parts[33].

The Wufoshan Group, which unconformably overlies both the Dengfeng Complex and Songshan Group, comprises a lower section (Ma'anshan Formation) of basal conglomerate, quartz sandstone, siltstone, and mudstone, with negligible metamorphism and deformation (Supplementary Fig. 2e–h). The sandstone is composed mainly of rounded detrital quartz grains that contain dust rims and quartz overgrowths (Supplementary Fig. 3i).

### Structural style

The Songshan Group is strongly deformed, with numerous fold and thrust structures developed at outcrop to regional scales[33]. At a regional scale, N- to NE-striking gentle to tight isoclinal folds developed due to E–W to NW–SE-directed shortening (Fig. 2c, Supplementary Fig. 5a). Most tight isoclinal folds are overturned, and the axial planes mainly dip to the NW. The folding is stronger in the central and western domains (Luohandong and Wuzhiling formations). At the outcrop scale, tight, chevron, recumbent, commonly rootless folds, asymmetric boudinage, S–C fabrics and duplex structures are widely developed in the lower Songshan Group (Fig. 4; Supplementary Fig. 5b–d). Collectively, the asymmetric structures indicate top-to-the-SE or -NE thrusting and shear sense at middle-to-upper crustal levels (Figs. 2d, 4). The overall structural patterns of the Songshan Group are broadly similar to those of modern fold-and-thrust belts, such as in the Alpine orogen.

### Detrital zircon U–Pb geochronology

To constrain the maximum depositional age, metamorphic age and provenance of the (meta)sedimentary rocks, we analysed zircon and/or rutile grains within samples of basal conglomerate, quartzite and quartz-mica schist from the Songshan Group, along with a quartz sandstone from the Ma'anshan Formation of the Wufoshan Group (Methods, Supplementary Data 1). Detailed results are presented in Supplementary Note, with U–Pb concordia diagrams and probability distribution diagrams shown in Supplementary Figs. 6 and 7, respectively.

Detrital zircons from the Luohandong and Wuzhiling formations of the Songshan Group have $^{207}Pb/^{206}Pb$ ages of 3.45–2.38 Ga (n = 213), and 2.78–2.33 Ga (n = 147), respectively. The Luohandong Formation samples contain three major zircon age populations at 3.45–3.35 Ga, 2.76–2.58 Ga, and 2.57–2.45 Ga, with a minor population at 2.44–2.38 Ga (Supplementary Fig. 7c). In contrast, the Wuzhiling Formation has a major age population at 2.56–2.45 Ga with a 2.51 Ga maximum, along with a subordinate older population at 2.77–2.57 Ga with a peak at 2.69 Ga (Supplementary Fig. 7g).

Detrital zircons in a quartz-mica schist sample from the upper Dengfeng Complex below the unconformity exhibit a single age peak at 2.56 Ga (Fig. 5a; Supplementary Fig. 7b). Detrital zircons in the quartz sandstone sample from the lower Wufoshan Group (Ma'anshan

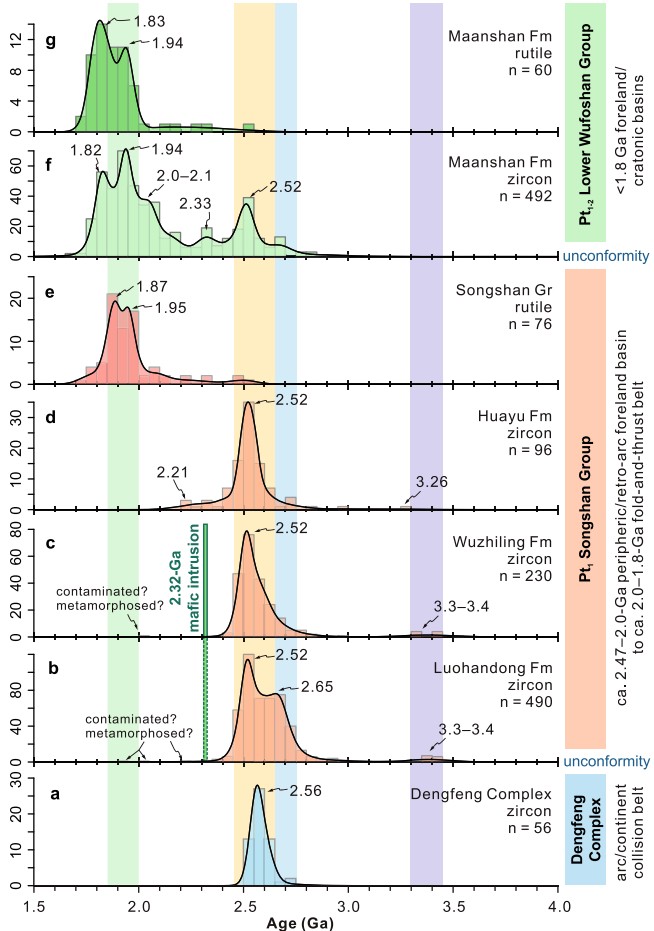

**Fig. 5 | Zircon and rutile U–Pb age patterns.** Kernel density estimates of detrital zircon and detrital/metamorphic rutile U–Pb ages from samples of the Dengfeng Complex (**a**), Songshan (**b**, **e**) and Wufoshan (**f**, **g**) groups. Note the different age patterns between rutile (**e**) and zircon (**b–d**) from the Songshan Group. Fm Formation, Gr Group.

Formation) yield a span of ages from 2.9 to 1.79 Ga, with dominant age populations at 2.67, 2.56, 2.45, 2.34, 2.14, and 2.0–1.80 Ga (Supplementary Fig. 7k). Notably, some detrital zircon grains have core–rim structures, in which metamorphic rims have ages between 2.0–1.9 Ga except for two >2.1 Ga (Supplementary Fig. 8a). Some Paleoproterozoic zircons (ca. 2.1–1.8 Ga), with or without rims, have low Lu contents and $(Lu/Dy)_N$ ratios (Fig. 6), showing depletion in heavy rare earth elements (HREE) in chondrite-normalised diagrams (Supplementary Fig. 8c).

## Rutile U–Pb geochronology

Rutile grains in the samples of quartzite and metaconglomerate from the lower Songshan Group are euhedral, subhedral, or irregular, and generally oriented (sub)parallel to the dominant foliation (Fig. 7a–c; Supplementary Fig. 6i–k). Analyses of rutile yield U–Pb ages of ca. 2.50 Ga ($n = 3$), 2.31–2.05 Ga ($n = 11$) and 2.00–1.75 Ga ($n = 62$), with peaks at 1.95 Ga and 1.87 Ga (Fig. 5e). The pressure-independent Zr-in-rutile (ZIR) geothermometer indicates temperatures of 469–586 °C (mean 505 °C) for <2.0 Ga rutile grains (Fig. 7e, g), which are consistent with metamorphic temperatures obtained by the Ti-in-biotite geothermometer (Supplementary Note, Data 2 and Tables 2, 3).

In contrast, rutile grains from the unmetamorphosed quartz sandstone of the lower Wufoshan Group are well-rounded (Fig. 7d), consistent with a detrital origin. They have U–Pb ages of 2.5–2.0 Ga ($n = 6$) and 1.98–1.75 Ga ($n = 54$), with two populations clustering at 1.94 Ga and 1.83 Ga (Fig. 5g). Rutile grains from the lower Wufoshan Group yield higher ZIR temperatures of 577–876 °C (except for one 479 °C), with a mean of 740 °C (Fig. 7f, g).

## $^{40}Ar/^{39}Ar$ geochronology

We obtained high-quality $^{40}Ar/^{39}Ar$ ages (Methods, Supplementary Data 3) for white mica and amphibole from metamorphic rocks of the central Dengfeng Complex known to record Neoarchean peak metamorphic ages and evidence for Paleoproterozoic overprinting[32,37]. Amphibole from the two different amphibolite samples yields $^{40}Ar/^{39}Ar$ plateau ages of 2035 ± 6 Ma (MSWD = 1.14, $p$ = 0.33) and 1861 ± 4 Ma (MSWD = 1.65, $p$ = 0.06) (Supplementary Fig. 9a, b). White mica grains from two garnet quartz-mica schist samples yield $^{40}Ar/^{39}Ar$ plateau ages of 1826 ± 2 Ma (MSWD = 0.45, $p$ = 0.92) and 1816 ± 3 Ma (MSWD = 1.7, $p$ = 0.16) (Supplementary Fig. 9c, d).

## Depositional and metamorphic ages of the Songshan foreland succession

Due to the absence of dateable volcanogenic horizons, the depositional age of the Songshan Group is imprecisely known, and can only be bracketed by the youngest reliable age of detrital zircon (maximum depositional ages, MDA) and that of the oldest magmatic intrusion (minimum depositional ages). After considering different approaches (see refs. 38,39, Methods), we use the weighted mean ages of the youngest cluster that overlaps in age with 1σ uncertainty (YC1σ) as a conservative estimate of MDA (Supplementary Fig. 10 and Table S1).

Detrital zircons from the metamorphosed basal conglomerate of the lower Songshan Group yield a YC1σ age of 2465 ± 22 Ma ($n = 5$). The two quartzite samples from the Luohandong Formation yield YC1σ ages of 2429 ± 19 Ma ($n = 4$) and 2511 ± 10 Ma ($n = 13$). The quartzite and quartz-mica schist samples from the Wuzhiling Formation yield YC1σ ages of 2490 ± 8 Ma ($n = 34$) and 2507 ± 6 Ma ($n = 35$), respectively. These data, combined with zircon U–Pb ages of 2.32–2.30 Ga[35,36] for mafic bodies intruded within the Wuzhiling Formation, constrain the depositional ages of the lowermost two formations of the Songshan Group to 2.47–2.32 Ga. A quartzite sample from the uppermost Songshan Group (Huayu Formation) yields a YC1σ age of 2241 ± 20 Ma ($n = 4$) and has a minimum depositional age of ca. 2.0–1.8 Ga based on the presence of metamorphic rutile (see below) that records post-depositional deformation. Detrital zircon and rutile from the Ma'anshan Formation of the lower Wufoshan Group yield YC1σ ages of 1813 ± 11 Ma ($n = 13$) and 1770 ± 10 Ma ($n = 10$), respectively, indicating an MDA of 1.77 Ga for the lowest Wufoshan Group.

The timing of metamorphism of the Songshan Group can be constrained by metamorphic rutile, which serves as a useful tool to date tectonothermal overprinting that is beyond the ability of zircon[40]. The young rutile grains (<2.0 Ga) from the Songshan Group are interpreted to be of metamorphic origin during post-depositional events based on the following lines of evidence: 1) the rutile grains exhibit a consistent alignment in the dominant foliation and have euhedral, subhedral and irregular shapes (Fig. 7a–c); 2) the peak ages of 1.95 Ga and 1.87 Ga are markedly younger than those of the detrital zircons (Fig. 5b–e); 3) the ZIR temperature of <2.0 Ga rutile grains is low (~469–586 °C) and comparable with the Ti-in-biotite temperature (~558–598 °C at 4 kbar) (Supplementary Tables 2–3); and 4) subordinate rutile grains with ages exceeding 2.0 Ga are inferred to have survived during prolonged Paleoproterozoic (at ca. 2.0–1.8 Ga) metamorphism, implying that the peak metamorphic temperature of the Songshan Group was lower than the closure temperature (~600 °C) of rutile in the U–Pb isotope system[41].

As the Songshan Group is unconformably overlain by the lower Wufoshan Group, cross-cut by 1.78-Ga post-kinematic granite, and contains metamorphic rutile, the timing of metamorphism and deformation of the Songshan Group is constrained at ca. 2.0–1.8 Ga. This is further substantiated by our samples from the underlying

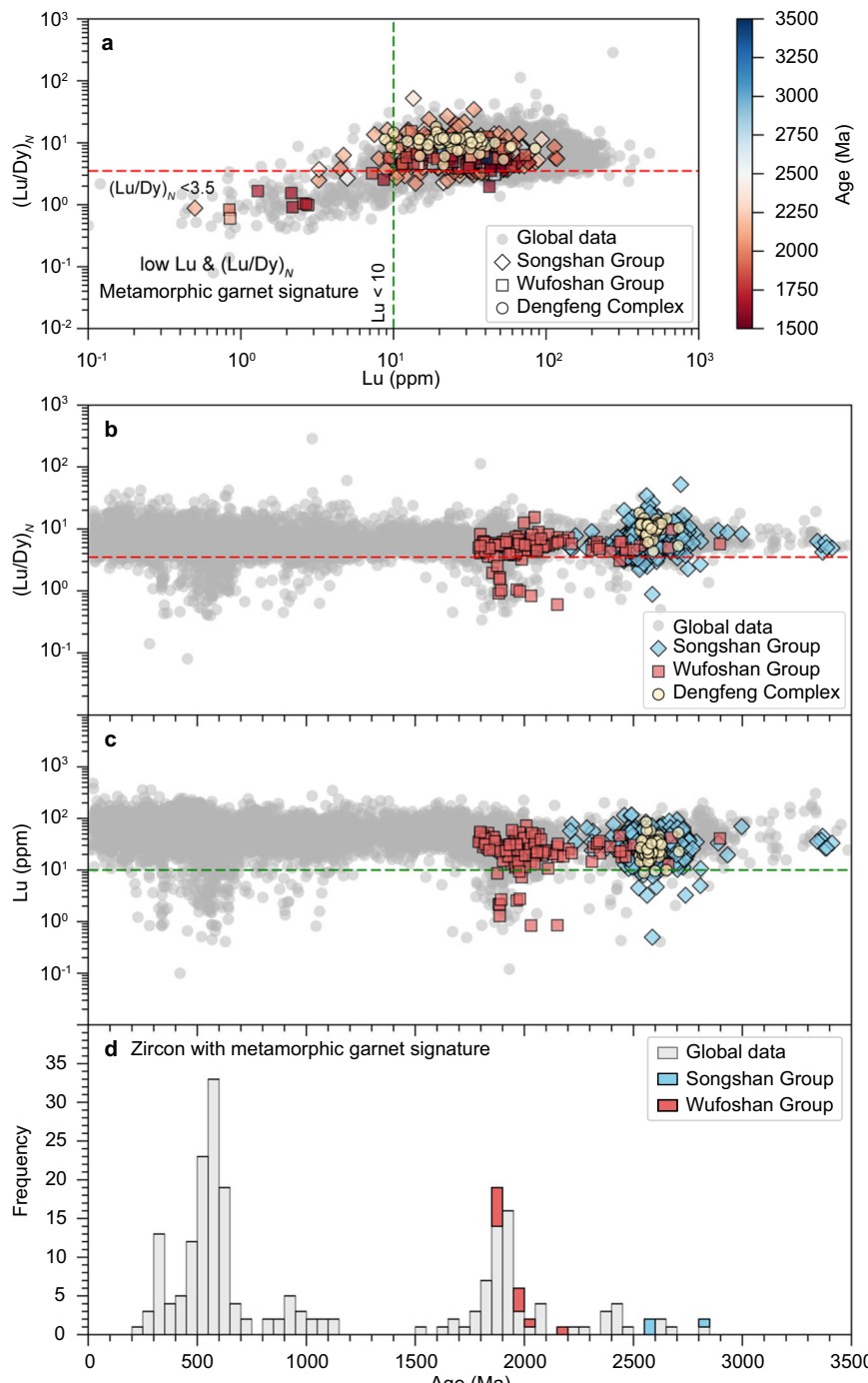

**Fig. 6 | Rare earth element characteristics of detrital zircon. a** $(Lu/Dy)_N$ vs Lu. **b** $(Lu/Dy)_N$ vs age. **c** Lu vs. age. **d** Stacked frequency histogram of zircon with low Lu and $(Lu/Dy)_N$. The zircon with low Lu and $(Lu/Dy)_N$ is regarded as a useful indicator of garnet (metamorphic garnet signature) during zircon growth, signifying high-pressure metamorphism. Subscript $N$ refers to chondrite normalised values. The global data set is from ref. 63.

Dengfeng Complex in which a coeval Paleoproterozoic (ca. 2.03–1.82 Ga) tectono-thermal overprint is recognised based on amphibole and mica $^{40}Ar/^{39}Ar$ dating (Supplementary Fig. 9).

## Sedimentary provenance in response to collisional orogenic events

Detrital zircon and rutile in sedimentary rocks provide key constraints on their igneous and/or metamorphic source regions, and hence the source-to-sink history[22,40,42]. In addition to our new data ($n = 745$), we compiled published zircon U–Pb data from the Songshan and Wufoshan groups that include only analyses with <5% age discordance

(Supplementary Data 4). All zircon ($n = 1364$) and rutile ($n = 136$) ages are plotted in Fig. 5.

All tectonic domains within the North China Craton have abundant 2.55–2.5 Ga basement rocks, but show differences in the proportions of older and younger rocks[43] (Fig. 2b). The (Proto-) Eastern Block contains several old continental fragments, with zircon U–Pb ages of 3.8 Ga (Anshan, East Hebei, Xinyang), 3.4–3.2 Ga (Anshan), 3.0–2.9 Ga (East Hebei), 2.92–2.72 Ga (e.g., Lushan, West Shandong, Zanhuang)[44,45]. The Central Orogenic Belt contains minor 2.70–2.65 Ga and 2.4–1.9 Ga igneous rocks[16,25,46], whereas the Western Block mainly comprises ca. 2.3–1.9 Ga igneous and sedimentary rocks (dominantly

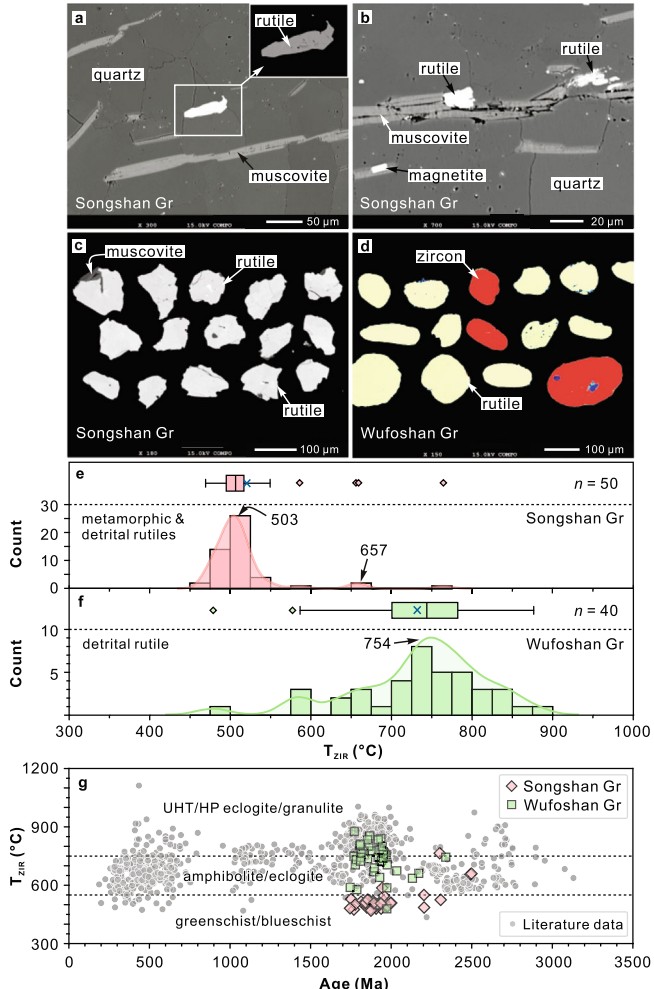

**Fig. 7 | Back-scattered electron (BSE) images and Zr-in-rutile temperatures.** **a**, **b** Inferred metamorphic in situ rutile grains aligned with the foliation of quartzite sample (20SS01) from the Songshan Group. **c** Separated rutile grains from the Songshan Group show euhedral, subhedral or irregular shapes. **d** Coloured BSE images showing well-rounded detrital rutile (and zircon) grains from quartz sandstone (20DF03) of the overlying lower Wufoshan Group, which are different from the metamorphic rutiles from the Songshan Group. **e**, **f** Histograms, kernel density estimations, and boxplots showing Zr-in-rutile temperatures ($T_{ZIR}$) for rutile from the Songshan Group and the lower Wufoshan Group. The dark blue cross represents the average temperature. **g** $T_{ZIR}$ versus age diagram. The literature data and metamorphic facies boundaries are from ref. 62 UHT ultra-high temperature, HP high pressure.

khondalite series, granitoids, and mafic intrusions)[47]. Notably, 1.95–1.80 Ga metamorphic events are widespread in the basement rocks of the North China Craton, notwithstanding several local exceptions that include the interior of the Eastern Block[25,48,49].

Detritus within the lower section of the Songshan Group (Luohandong and Wuzhiling formations) contains diagnostic 3.4–3.2 Ga and 3.00–2.85 Ga zircon populations that are characteristics of the Eastern Block, whereas the dominant 2.7–2.5 Ga zircons were likely derived from both the Eastern Block and central orogen, as the Western Block was likely separated from these two domains in the early Paleoproterozoic[26]. Multiple lines of evidence, including Neoarchean (ca. 2.56–2.50 Ga) arc magmatism[50], mélanges[51,52], and paired metamorphism[32], suggest that the Central Orogenic Belt represents a Neoarchean to earliest Paleoproterozoic accretionary-to-collisional orogen between intra-oceanic arc systems to the west and the Eastern Block to the east[25]. The arc(s) and frontal accretionary complexes were

thrust upon continental margin sequences along the western margin of the Eastern Block[16,51,53,54]. The assembled orogenic wedge and continental block were unconformably overlapped by the lowest Songshan foreland basin sequences (<2.47 Ga), which received large volumes of detritus from the growing orogenic core and cratonic interior. The upper section of the Songshan Group records an increased contribution from 2.31–2.20 Ga detritus likely sourced from middle-Paleoproterozoic (mostly Rhyacian) magmatic rocks in the central orogen[46].

The overlying lower Wufoshan Group contains magmatic zircon age populations at 2.52 Ga and 2.33–2.10 Ga, and with age peaks of magmatic and metamorphic zircons at 1.94 Ga and 1.82 Ga (Fig. 5f). The age peaks of detrital rutile at 1.94 Ga and 1.83 Ga (Fig. 5g) indicate that their sources underwent significant Paleoproterozoic (Orosirian) metamorphism. Collectively, our data are consistent with a detrital source dominated by thickened orogenic crust along the central orogen of the North China Craton where Paleoproterozoic (ca. 2.3–2.1 Ga) magmatic rocks are abundant[46,55]. The high-grade metamorphism recorded by detritus of rutile and metamorphic zircon (Supplementary Fig. 8) was related to the mid-Paleoproterozoic collisional orogenesis and cratonisation[26].

## Depositional-to-deformational evolution records two styles of orogenesis in North China

Our new data for the Songshan Group provide important constraints on Archaean to Paleoproterozoic orogenesis during the Precambrian evolution of the North China Craton (Fig. 8). The lowest Songshan Group (2.47–2.32 Ga) represents an early Paleoproterozoic foreland basin succession that unconformably overlapped the orogenic wedge and cratonic margin of the (Proto-)Eastern Block due to 2.5–2.47 Ga arc–microcontinent collision[25], providing the key sedimentary evidence for terrane assembly and orogenesis at the end-Archaean to early Paleoproterozoic (Fig. 8a). Arc–microcontinent collision was likely followed by a reversal of subduction polarity[56], leading to eastward subduction of oceanic lithosphere beneath the western margin of the Eastern Block that converted to an active continental margin. The upper Songshan Group (ca. 2.24–2.0 Ga) evolved into either a retro-arc foreland basin[28], retro-arc extensional basin[25], or mid-Paleoproterozoic (ca. 2.3–2.0 Ga) rifting/cratonic margin basin[57] during the Paleoproterozoic, the details of which require further investigations. Nevertheless, the sedimentary basins and underlying Archaean basement of the North China Craton underwent strong modification (e.g., far-field effect) during late Paleoproterozoic (ca. 2.0–1.85 Ga) orogenesis, consistent with several Paleoproterozoic orogenic belts along its northern margin[25,49] and the interior[26,27,57,58].

The former Songshan foreland sequence was located in the front of the orogen, becoming a fold-and-thrust belt during the late Paleoproterozoic, as revealed by the strong compressional deformation and ages of metamorphic rutile (Fig. 8b). The underlying basement of the Dengfeng Complex records $^{40}Ar/^{39}Ar$ ages of 2.03–1.82 Ga (Supplementary Fig. 9), reflecting Paleoproterozoic tectono-thermal overprinting of the former Neoarchean orogenic wedge[32]. Thus, the Songshan Group records the life-cycle from an early peripheral foreland basin (ca. 2.47 Ga) to a retro-arc foreland basin, and finally to a fold-and-thrust belt (ca. 2.0–1.8 Ga), which was then unconformably overlapped by the younger foreland or intra-cratonic succession of the lower Wufoshan Group (<1.77 Ga) (Fig. 8c). These sedimentation-to-deformation processes were in response to two episodes of orogenesis in the North China Craton, and provide a partial record of the assembly of the Neoarchean supercratons[59,60] (at ca. 2.5 Ga) and the amalgamation of the Paleoproterozoic Columbia supercontinent at 2.0–1.8 Ga[9,61].

The majority of rutile in metamorphic rocks has been suggested to form under relatively high pressure[62]. Rutile preserved in sedimentary rocks can therefore be used to characterise any potential change in metamorphism of the detrital source. However, the possible

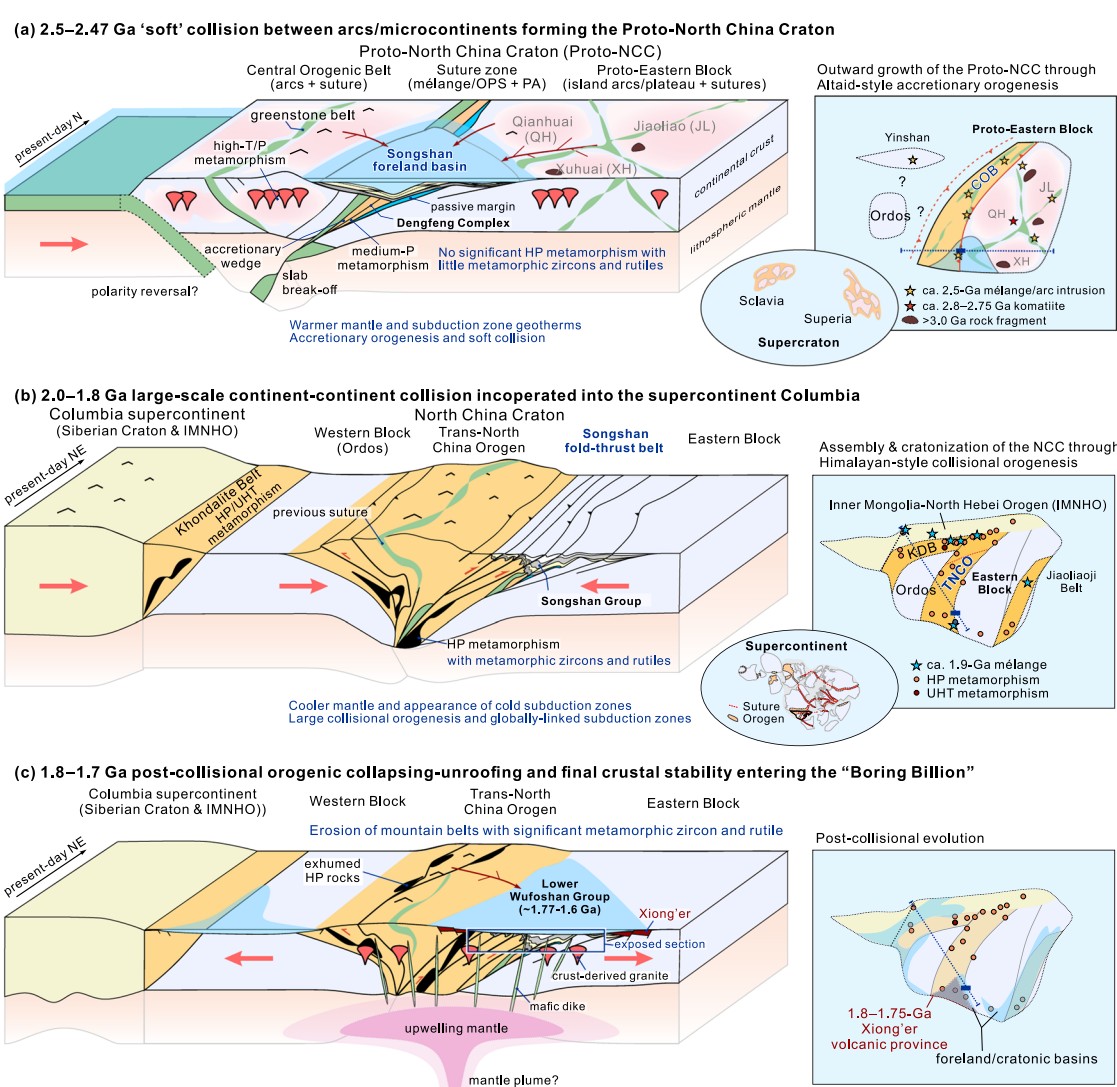

**Fig. 8 | Tectonic model for the two episodes of orogenesis and their geological response in the North China Craton. a** Circa 2.5–2.47 Ga, Altaid-style 'soft' collision between arc(s)/micro-continent(s) forming the Proto-North China Craton that consists of the central arc(s) and micro-continents/arcs/oceanic plateaus in the eastern domain of the North China Craton (Proto-Eastern Block)[16,24,25,93]. The closure of oceans and terrane assembly is evident by the occurrence of mélanges and accretionary complexes, Paleoproterozoic unconformities (ca. 2.47 Ga), and post-kinematic potassic granitoids (ca. 2.48–2.42 Ga). The processes are similar to the terrane accretion and 'soft' collision in more-recent accretionary orogens, such as the Altaids[65,94], which is characterised by intra-oceanic subduction and scarcity of exhumation of high-pressure (HP) rocks in subduction-collision zones.
**b** 2.0–1.80 Ga, large-scale continent-continent collisional orogenesis led to the

cratonisation of the North China Craton and its incorporation into the supercontinent Columbia[25,73], which formed Himalayan-style collisional orogenic belts[72,83] and orogenic plateaus[24,25]. During the collision, the hinterland of the orogenic crust was buried to a deep-crustal level and underwent high-pressure metamorphism, forming HP granulite/eclogite metamorphic domains that contain metamorphic zircon, garnet, and rutile, while in the orogenic margin, the basement rocks including the former Songshan Group foreland sequence, were strongly contracted and deformed, forming a fold-and-thrust belt. **c** 1.8–1.7 Ga, post-orogenic extensional collapse and unroofing resulted in the development of the Xiong'er igneous province and subsequent widespread stable intra-cratonic or craton margin basins, entering the 'Boring Billion'.

---

preservation bias of rutile (e.g., Archaean) should be evaluated before interpreting the detrital rutile data set, as older rutile could be recrystallised or reset by younger higher-grade metamorphism due to its relatively low U–Pb closure temperature (~500–600 °C)[42]. But our results, including textures, ages and Zr-in-rutile temperatures, provide compelling evidence that ca. 2.0–1.8 Ga rutile grains within the majority of the Songshan Group were neocrystallised during the Paleoproterozoic metamorphism rather than being recrystallised or reset from older Archaean detrital rutile grains. Although rutile from the Songshan and Wufoshan groups records similar U–Pb age patterns (Fig. 5e, g), it has different origins and metamorphic temperatures (Fig. 7e, f). Most rutile from the lower Songshan Group is of metamorphic origin (mean ZIR temperature = 505 °C) and records Orosirian

ages (ca. 2.0–1.8 Ga, *n* = 62) for post-depositional tectonothermal overprinting, with minor Neoarchean detrital rutiles (ca. 2.50 Ga, *n* = 3) derived from the source regions (Fig. 7, Supplementary Table 2). In contrast, rutile from the overlying unmetamorphosed Wufoshan Group is of detrital origin, and records much higher-grade Paleoproterozoic metamorphism (mean ZIR temperature = 740 °C for 2.0–1.8 Ga rutile) in the source regions. Therefore, minor ca. 2.50 Ga detrital rutile grains in the lower Songshan Group that contrast with the abundant 2.0–1.8 Ga detrital rutile in the Wufoshan Group can be used to constrain the distinctive metamorphic characteristics during the Neoarchean and the Paleoproterozoic. Additionally, detrital zircon from the lower Wufoshan Group exhibits significant core–rim structures (~21%), with rims dominated by Paleoproterozoic ages

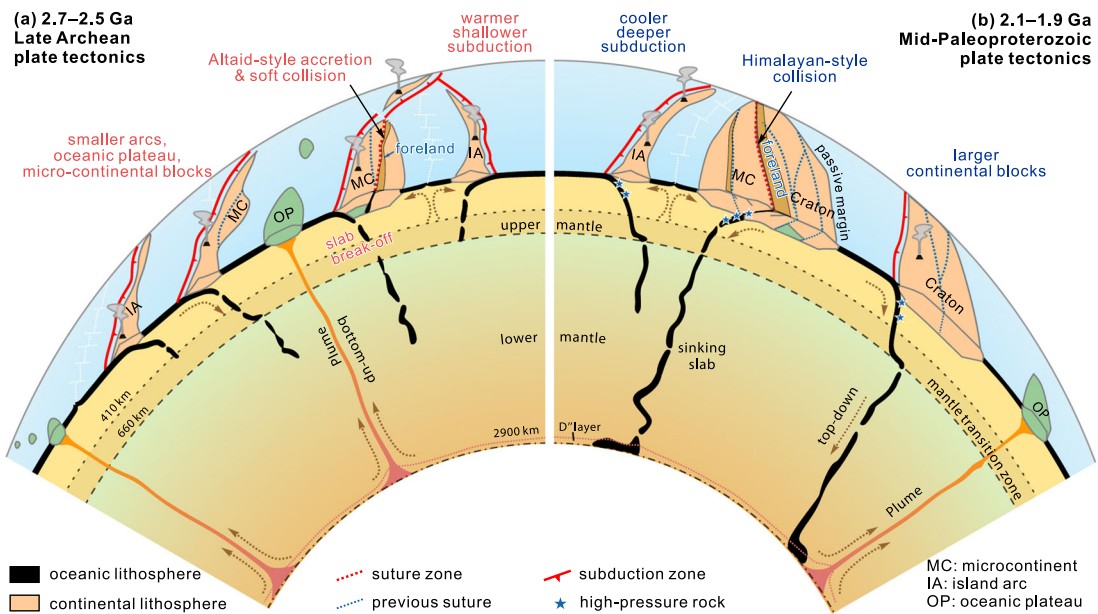

**Fig. 9 | Schematic model showing the two styles of plate tectonics and orogenesis in the late Archaean (2.7–2.5 Ga) and middle Paleoproterozoic (ca. 2.1–1.9 Ga). a** Late Archaean plate tectonics was characterised by generally warmer and shallower subduction zones and probably more frequent slab break-off due to a warmer mantle, which led to vertical crustal growth in juvenile island arcs through intense slab melting and lateral accretion by Altaid-style 'soft' collision between arcs and/or micro-continents[16], forming several supercratons at the end of the Archaean[60]. **b** Mid-Paleoproterozoic plate tectonics was characterised by a more frequent occurrence of deeper and cooler subduction, and global Himalayan-style collisional orogens between cratons and/or large continental blocks, resulting in the formation of the supercontinent Columbia[8,10]. Orogenesis and associated tectonic denudation and continental weathering played an important role in shaping Earth's near-surface environments, which in turn influenced the subduction-collision processes.

(Supplementary Fig. 8a). Furthermore, some zircons have HREE-depleted signatures with low Lu contents and $(Lu/Dy)_N$ ratios (Fig. 6), implying they have a metamorphic garnet signature formed in a high-pressure (HP) setting[63]. These rutile and zircon provenance data together reflect two different metamorphic patterns: the first during the ca. 2.50–2.47 Ga arc–microcontinent collision and the second during Paleoproterozoic continent–continent collision, likely attributed to different orogenic styles.

In the late Neoarchean, small-scale arc–arc or arc–microcontinent 'soft' collisions led to the outward growth around existing cratonic nuclei[16,25], forming large proto-cratonic continental landmasses (e.g., possible supercratons) (Fig. 8a). The style of such arc and/or microcontinent accretion and soft collision is analogous to Phanerozoic accretionary orogens, such as the Altaids (Central Asian Orogenic Belt), where most accretion and growth of Phanerozoic juvenile crust occurred[64–66]. Warm subduction with subsequent soft accretion-collision may be distinctive and characteristic of a warmer thermal state of the lithosphere and underlying mantle, which enabled intense slab melting and crustal growth[16,67,68], but hindered the generation of abundant low-temperature–high-pressure metamorphic rocks in subduction-collision zones[69]. Consequently, rutile and HREE-depleted zircons are relatively rare in exhumed crustal metamorphic rocks, consistent with the detrital archive[62,63] (Figs. 1a, 6, 7).

In the late Paleoproterozoic, the widespread occurrence of metamorphic rutile (Fig. 7g) and zircon (Supplementary Fig. 8a–c), as recorded by sedimentary rocks of the Wufoshan Group, indicate intense metamorphism at generally higher pressures. This is supported by the occurrence of high-pressure rocks (e.g., HP granulites, retrograde eclogites, low-$T$ eclogite xenoliths[70]) in the North China Craton[27,48,71], which is attributed to cold subduction and continent–continent collision during cratonisation of the North China Craton (Fig. 8b). The style (e.g., magmatism, metamorphism, sedimentation, scale and configuration) of Paleoproterozoic orogenies has been regarded as very similar to modern collisional orogenesis, such as

the Himalayan orogen[25,58,72]. Furthermore, nearly all cratons record significant orogenic events at this time, which have been linked with the formation of the supercontinent Columbia (Fig. 2a)[9,73]. The lower Wufoshan Group that unconformably overlapped Archaean and early Paleoproterozoic basement in the North China Craton was likely deposited in a foreland or cratonic margin basin adjacent to a late Paleoproterozoic orogenic belt (Fig. 8c), receiving abundant detritus, including 2.0–1.8 Ga rutile and metamorphic zircon from the hinterland due to unroofing and erosion of the exhumed deep-level orogenic crust.

## Linking orogenesis with the dramatic change in near-surface environments

The significant shift in orogenic styles as recorded by foreland basins in the North China Craton, albeit occurring at several times from the Neoarchean to the late Paleoproterozoic (ca. 2.7–1.8 Ga), is also observed in other cratons, including those in India, North America, Australia, and South Africa[74–76]. These changes likely had profound impacts on Earth's near-surface environments, including the land, hydrosphere, atmosphere and biosphere[1,16].

Late Archaean cratonisation was associated with Altaid-style arc–arc and/or arc–microcontinent accretion and soft collision, leading to the outward growth of continental crust and rapid emergence of subaerial landmasses[5,16,25] (Fig. 9a). The exact elevation might have been comparable with modern arc–continent collisions, as in the Western Pacific accretionary systems. The emergent landmasses and increasing elevation enabled intense tectonic denudation and silicate weathering, resulting in an increase of nutrients (e.g., P, Mo) to the ocean and, consequently, the development of marine primary productivity and a concomitant increase in the level of atmospheric oxygen during the early Paleoproterozoic (2.45–2.1 Ga, Fig. 1c), known as the GOE[7]. Such change in weathering rates is recorded in seawater $^{87}Sr/^{86}Sr$ ratios, which increased rapidly from ca. 2.5 Ga and reached a peak at 2.3–2.2 Ga (Fig. 1b)[77]. Thick clastic (quartzite-dominated) units

in the Songshan Group foreland sequences are consistent with emergent continental landmasses and high rates of weathering and erosion. The presence of GIF, carbonates, and purple quartzites and schists in the middle-upper parts of the succession indicate oxidative weathering[35].

Detrital zircons from the lower Songshan Group were mainly derived from felsic sources, presumably dominated by TTG gneisses and volcanic rocks in greenstone belts. The dominance of mature (quartz-rich) sedimentary rocks requires the dissolution of huge proportions of the feldspar and other silicate minerals, which must have consumed large volumes of $CO_2$[19,78]. The liberated $Ca^{2+}$ and $Mg^{2+}$ cations led to widespread deposition of marine carbonate rocks worldwide, which may have played an important role in carbon sequestration[19]. In addition, a lower rate of volcanism implied by the global magmatic slowdown (or 'tectono-magmatic lull'[79]) at 2.4–2.2 Ga inhibited (eruptive) recycling of $CO_2$ back into the atmosphere. The imbalance between the source and sink of carbon induced by these tectonic and sedimentary processes likely caused the decrease in atmospheric $CO_2$ and 2.4–2.1 Ga icehouse conditions (Huronian glaciation)[11] (Fig. 1c).

At ca. 2.2–2.0 Ga, carbon- and phosphorus-rich sediments such as graphite-bearing schists and phosphorites appear widely in the North China Craton (Fig. 1d) and other cratons[55,80], consistent with the high marine primary productivity and increased burial of organic carbon as evidenced by a large positive $\delta^{13}C$ excursion (the Lomagundi event, Fig. 1b). These changes have been linked to partial contributions from the Orosirian large igneous provinces and continental weathering following supercraton rifting[55]. The enhanced rates of weathering, along with thawing of the snowball Earth, would have provided more sediments to the trenches, which would have lubricated subduction zones[1,81] and promoted cold and deeper subduction between ca. 2.1–1.9 Ga, consistent with the local appearance of high-pressure metamorphic rocks[82]. Furthermore, the increased carbon-rich sedimentary rocks would lead to reduced frictional strength, which promoted crustal thickening in the Paleoproterozoic mountain belts[80].

Mid-Paleoproterozoic accretion-to-collision orogens that developed between and at the margins of cratons and/or large continental blocks (Fig. 9b) resulted in the formation of the supercontinent Columbia, which would have contained many large Himalayan-style orogenic belts and plateaus[10,63,72,83], as recorded by higher peak metamorphic pressures (Fig. 1a) and the occurrence of HP eclogites and collisional orogenic belts worldwide[8,70,71,83]. Deeper and cooler subduction and enhanced collisional orogenesis can be attributed to higher lithospheric strength that is a direct consequence of cooling of the mantle and the emergence of orogenic (plate tectonic) styles more akin to the modern style (Fig. 9b). During collisional orogenesis, the hinterland of the orogenic crust underwent high-pressure metamorphism to form HP granulite and eclogite facies metamorphic belts with abundant metamorphic zircon, garnet, and rutile[26,70]. In the orogenic forelands, the basement rocks were strongly contracted and deformed to form large-scale fold-and-thrust belts[58], as exemplified by the deformation of the Songshan foreland sequence.

The long-term uplift and unroofing of Himalayan-style orogens during the late Paleoproterozoic led to strong tectonic denudation and silicate weathering of evolved continent crust, as evidenced by detrital zircon and rutile in the foreland/cratonic basins, such as the lower Wufoshan Group (<1.77 Ga) reported here, and the significant increase in continent-derived seawater Sr signal between 2.0–1.7 Ga (Fig. 1b). The orogens were largely denuded by the onset of the eponymous 'Boring Billion' (1.75–0.75 Ga)[84]. Younger (<1.8 Ga) foreland, intra-cratonic and cratonic platform sequences, including the lower Wufoshan Group (Fig. 8c), record the stabilisation and final cratonisation of major cratons worldwide[57] that paved the way for the appearance of the oldest known eukaryotes[12] (Fig. 1c, d).

In conclusion, this study provides new constraints on the depositional-to-deformational evolution of the Paleoproterozoic Songshan Group foreland sequence, one of the oldest documented orogenic foreland basins in the North China Craton, from sedimentary and deformational perspectives. Importantly, it underscores the significance of multi-mineral geochronological analysis of key minerals from sedimentary archives in ancient orogenic forelands, which provide clues as to how sedimentation and metamorphism responded to changing orogenic style. Our research has shed light on changes in the orogenic styles during the Neoarchean to Paleoproterozoic and their possible influence on global biogeochemical cycles, which offers fresh perspectives on the dynamic interplay between solid and surficial Earth processes, contributing to a deeper understanding of our planet's evolution towards a more habitable environment.

## Methods

### Zircon and rutile U–Pb isotopic analyses

Zircon U–Pb dating was conducted at the Hubei Geological Research Laboratory, Ministry of Land and Resources of China, Wuhan. Cathodoluminescence (CL) images of zircons were obtained by a Gatan Mono CL4 + CL system attached to a Zeiss Sigma300 field emission scanning electron microscope. Zircon U–Pb dating was performed using LA-ICP-MS utilising a GeoLas Pro 193 nm excimer ArF laser ablation system (Coherent Inc., Göttingen, Germany) attached to an Agilent 7700x ICP-MS. The spot diameter was 32 μm. Zircon 91500 was analysed twice every five analyses as an external standard, and zircons GJ-1 and Plešovice were measured as monitoring standards. All zircon standards yield ages consistent with the recommended values within error (see details in Supplementary Data 1). The synthetic silicate glass NIST SRM 610 was used as the standard to calibrate the trace element compositions.

Rutile U–Pb dating and trace element analysis were conducted at the State Key Laboratory of Geological Processes and Mineral Resources, China University of Geosciences, Wuhan, by LA-ICP-MS utilising a GeoLas HD 193 nm excimer ArF laser ablation system (Coherent Inc., Göttingen, Germany) and an Agilent 7900 ICP-MS. All data were acquired by single spot ablation at a spot size of 44 μm. Each measurement consisted of 20 s of acquisition of the background signal followed by 50 s of ablation signal acquisition. Rutile standard R632 was analysed twice every eight analyses as an external age standard, and rutile RMJG was measured as a monitoring standard. All standard ages were consistent with the recommended reference values (see details in Supplementary Data 1). The NIST 610 was used to calibrate trace elements. Off-line selection and integration of background and analyte signals, time-drift correction and quantitative calibration of trace element analyses, and data reduction followed the protocols given by refs. 16,85. Zircon and rutile U–Pb isotope and trace element results are presented in Supplementary Data 1.

### $^{40}Ar/^{39}Ar$ geochronological analysis

The amphibole and mica $^{40}Ar/^{39}Ar$ analyses were conducted at the Western Australian Argon Isotope Facility at Curtin University, Australia. The detailed analytical procedure and the relative abundance of Ar isotopic data are provided in Supplementary Note and Supplementary Data 3, respectively. The $^{40}Ar/^{39}Ar$ plateau age diagram is shown in Supplementary Fig. 9.

### Estimate of maximum depositional ages

The maximum depositional ages (MDA) of sedimentary rocks can be estimated based on different methods. The widely used methods include the youngest single grain (YSG), youngest cluster overlapping with 1σ or 2σ uncertainties (YC1σ, YC2σ), and youngest statistical peak (YSP)[38,39]. For early Precambrian detrital zircons, the age of the youngest single grain is not appropriate because of measurement uncertainties, potential Pb loss and later hydrothermal modification[86].

The ±2% analysis error of LA-ICP-MS zircon U–Pb dating would have a significant influence on the resulting measured ages for old zircons (e.g., ±50 Ma uncertainty for a 2500 Ma zircon). Thus, for the early Precambrian sedimentary rocks without significant Pb loss, the weighted mean age of the youngest peak (YSP) or youngest cluster overlapping with 1σ (YC1σ) or 2σ uncertainties (YC2σ) would be the most appropriate and robust estimate of MDA (~98% confidence). In this study, we use the YC1σ as the estimate of the MDA (Supplementary Table S1). The calculation of MDA was conducted using the Python-based code, detritalpy[87].

## Time series analysis

Time series regression analysis was conducted on the global datasets of thermobaric ratios (*T/P*) and peak pressure (*P*) of global metamorphic rocks during crustal metamorphism. The metamorphic data set is based on refs. 8,88, with the addition of recently reported Paleoproterozoic HP metamorphic rocks. We smoothed the data with a LOESS (locally weighted scatterplot smoothing) regression method using the bootstrap function in Acycle v2.6[89]. The number of bootstrap samplings was 1000. The shaded envelopes in corresponding curves indicate the range of 1σ and 2σ uncertainty (Fig. 1a, b). The LOESS regression can create statistically robust curves that fit through the data to assess general trends of crustal metamorphism and isotopes on the planetary scale.

## Data availability

All data used in this study are available in the Supplementary Information, Figshare (https://doi.org/10.6084/m9.figshare.24598434), and cited literature.

## Code availability

The code detritalpy is available from https://pypi.org/project/detritalpy/. The code Acycle is available from https://acycle.org.

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

## Acknowledgements

This research was supported by grants from the National Natural Science Foundation of China (41888101, T.M.K., B.H.; 41890834, T.M.K.; 42330104, D.F.; 42102244, B.H.), the China Postdoctoral Science Foundation (2021M692977, B.H.), 111 Project (BP0719022, T.M.K.), the MOST Special Fund (MSFGPMR2022-7, T.M.K., B.H.) of the State Key Laboratory of Geological Processes and Mineral Resources and the Fundamental Research Funds for National Universities (CUG2106365, B.H.; 2023075, D.F.) from the China University of Geosciences, Wuhan. T.E.J. acknowledges funding from the Australian Government through an Australian Research Council Discovery Project (DP200101104, T.E.J.). This is a contribution to the International Lithosphere Programme 'Formation, Character, History, and Behavior of Earth's Oldest Lithospheres' by the International Union of Geological Sciences (IUGS). B.H. thank Junpeng Wang, Fred Jourdan, Tao Luo and Shiyang Pan for their help during analytical experiments.

## Author contributions

B.H. conceived the research and wrote the original draft. B.H., D.F., M.L. and Q.Q. carried out field investigations, sampling, and experimental analysis. T.M.K. contributed to the conceptualisation and discussion about the foreland basin deposition. S.A.W. provided support for Ar-Ar analysis. D.F., T.M.K., T.E.J., S.A.W., H.D., M.L. and Q.Q. contributed to discussions, editing, and revisions.

## Competing interests

The authors declare no competing interests.
