## [Peer Review File · Nature Communications]

REVIEWER COMMENTS

Reviewer #1 (Remarks to the Author):

During the late Archean to Paleoproterozoic, the Earth's deep and surficial reservoirs underwent dramatic changes, as evidenced by a number of indications, which are essential to comprehending the variation and interaction of Earth's solid and surface systems. Nevertheless, the transformative mechanism by which plate subduction and plate boundary evolved at the critical Archean to Paleoproterozoic juncture still remains enigmatic. This contribution reports the identification and life cycle of an ancient Paleoproterozoic foreland basin in the North China Craton, one of the oldest foreland successions in the world, and highlights the long-term evolution from ca. 2.5-2.47 Ga deposition initiated by warm subduction and Altaid-style "soft" micro-continental collision and deformation-uplift-denudation due to ca 1.97-1.85 Ga Himalayan-style continental collision. This study adds to our understanding of Earth's early evolution by shedding light on the multiple stages of source-and-sink history of the foreland sediments in response to late Archean to Paleoproterozoic orogenic episodes as well as potential influences on the dramatic variations of surficial processes and conditions on early Earth. Despite that, there are still some concerns with this contribution, prompting me to recommend a minor revision prior to publishing.

Major concerns:

(1) Priority one is the connection between the tectonic evolution of the two episodes of orogenesis in North China Craton. Their respective tectonic evolution has been clearly manifested by the convincing works of the authors' group in recent years, however, the successive evolution history in the west of the "Proto-Eastern Block" (Figure 7A) is, in my viewpoint, an important element for constructing the tectonic architecture from "supercraton" to "supercontinent" over a timeframe of ~500 million years (i.e., from ~2.5 Ga to ~1.98 Ga) (Figures 7A-B). In any case, an ancient oceanic basin lay between the Proto-Eastern Block and Ordos Block. If this ocean was consumed by subduction zone along the western margin of the Proto-Eastern Block, (i.e., the COB transferred to an active margin), is there any geological (magmatic, metamorphic, or accretionary) record for the successive prolonged subduction from the late Archean to Paleoproterozoic? Alternatively, if the COB was indeed a passive margin, were sediments equivalent to the passive boundary deposited and preserved? In other words, the tectonic scenario between two episodes of orogenesis in the Archean and Proterozoic is likely a crucial link for elucidating the transformative mechanism for the systematic changes from deep to surface in early Earth.

(2) The structural style of the foreland deposits matches with the characteristics of the thrust-fold belts at the middle-to-upper crustal-level. Although rutiles have been utilized to constrain the metamorphic temperature (T) for their presumed sources, it is necessary to directly quantify the metamorphic T for the schists in the thrust-fold belts and to, on the other hand, mutually verify the provenance of the detrital rutiles.

(3) There must be some clerical errors in the text depicting the age peaks for the geochronologic spectra (Figure 6). For instance in line 173, one of the age peaks for detrital rutile is written as "1.87 Ga" but is shown as "1.86 Ga" in Figure 6E; in line 247, age peaks for detrital zircons are written as "1.93 Ga and

1.84 Ga” but are shown as “1.94 Ga and 1.83 Ga” in Figure 6G; etc. Please check thoroughly the consistency of the figure and text for the zircon and rutile U-Pb geochronology.

A few specific comments:

- (1) Lines 34-35, change “one of most...” to “one of the most...”;
- (2) Line 246, change “(Fig. 6G)” to “(Fig. 6F)”;
- (3) Figures 3A and 3B should have photographic scale.

Reviewer #2 (Remarks to the Author):

After carefully reviewing this submission, I don't think it can be published in NC, I recommend reject, the main reasons for my suggestion are as follows.

1. It seems that this study is based on a relatively small number of samples - especially given that you are trying to resolve large scale issues relating to the temporal development of the Paleoproterozoic foreland successions. In some cases, I found it difficult to actually determine how many samples had been used. It is imperative that you provide a clear statement regarding the exact number of samples used for each type of analysis - it is absolutely critical that it you can demonstrate that you have used sufficient samples to justify the conclusions that you have reached.

2. This study has no novelty and too local, the methods are simple, many publications had defined the two-stages orogenesis model of the North China Craton, Archean mélange and Paleoproterozoic foreland successions developed within the Trans-North China Orogen (see Professor M. Santosh's group papers, Professor Guo-chun Zhao's group papers, Professor Ming-guo Zhai's group papers, Professor T. Kusky's group papers, etc.)

3. Many concepts and descriptions are confused in the manuscript, for example, Nuna is different to Columbia supercontinent, see JG Meert et al. 2012 Gondwana Research; JG Meert and M Santosh 2017 Gondwana Research. Also, many Chinese strata names confused me, didn't explain them clearly. The citations are not rigorous in the manuscript, who was the first to propose this concept of Columbia supercontinent, the evolution models of Columbia supercontinent and North China Craton??? The authors even didn't add any references in many paragraphs/sentences, see lines 323, 360-367, etc.

4. The manuscript mixed use of full name and abbreviations here and there. For example, North China Craton and NCC; Circa and Ca.; Formation and Fm. (See Lines 52, 110, 123, 127-128, 140, 158, 201, 206, 317, 344, 661, 631).

5. The logic of this manuscript is weak, this work can't better define the differences and controversies among the models. The maximum depositional ages of the detrital zircons can't be represented the depositional age of the strata, not to mention such old metamorphic strata; The age analysis lacks evidence, you can't say pb-loss casually. The authors can't discuss more contributions about the controversies among the models of lines 260-267 and can't get this conclusion of lines 288-290 without the evidence.

6. I found the conclusions to be somewhat weak - they really do not stress the main contributions of the paper. The Conclusions need to be improved.

7. Comments on the figures:

Figure 1. Nuna or Columbia??? The timing of supercontinents are right?

Figures 2-5. the authors discussed the structures, not even providing the nature of faults in the Figure 2; the field interpretations seem unclear in the Figures 3-5; the authors should define the abbreviations in the figures. No any field data on the foliations and lineations in the Figure 3. Confused the COB and TNCO.

Figure 6. How does the author distinguish between the detrital/metamorphic ages ?

Figure 7. Too general and it is difficult to compelling to the readers because many places lack evidence, especially the time limits; also confused the COB and TNCO. It seems that the authors just want to mix everything together and doesn't clarify geological history very well. This manuscript does not contribute in terms of time constraints, it is too broad.

Reviewer #3 (Remarks to the Author):

Huang and colleagues present a detailed study of the structural history, basin evolution, and provenance of Paleoproterozoic sedimentary rocks in the North China Craton. They integrate these datasets to infer a contrast in the tectonic style of orogenesis and continental assembly in the North China Craton (NCC) between the late Neoproterozoic and early Paleoproterozoic. The paper concludes by discussing the

possible influences the purported transition in global tectonic regimes across the Archean-Proterozoic may have had on the wider Earth system.

Linking solid Earth processes and tectonic regimes to the wider Earth system is a highly active area of research and of interest to a broad cross section of the scientific community. Multi-disciplinary studies such the one presented in this paper are essential to developing a holistic understanding of relationship between tectonic processes and surface environments through deep time. The paper by Huang and colleagues takes the novel approach of studying the history of foreland sedimentary basins to document changes in tectonic regimes across the Archean-Proterozoic transition and its impact on global geochemical cycles. The paper is mostly well-written and is exceptionally and informatively illustrated. In my opinion, this paper would make an excellent contribution to Nature Communications. However, I have a number of queries regarding the provenance data that should be addressed before publication.

Detrital Rutile evidence

One of the main pieces of evidence cited for a change in orogenic style from hot 'soft' collision in the Neoproterozoic to hard 'cold' collision in the Paleoproterozoic is the near absence of Neoproterozoic detrital rutile in the Songshan Group contrasting with the abundance of Paleoproterozoic detrital rutile grains in the Wufoshan Group. This interpretation is consistent with the tendency of rutile to grow in high-P metamorphic rocks, which are more likely to be exhumed and eroded into adjacent sedimentary basins in 'colder' orogens. Some important caveats to this interpretation are the tendency of rutile to recrystallise to ilmenite or titanite during the greenschist- to amphibole facies of a prograde metamorphic path (e.g., Ashley and Law, 2015, *contrib. Mineral. Petrol.*, 169) and the relatively low closure temperature of Pb diffusion in rutile (~500-600 C = greenschist-amphibolite facies conditions).

The fact that the Songshan Group has been metamorphosed to greenschist facies raises the possibility that Neoproterozoic detrital rutile grains were originally present in these sedimentary rocks but were subsequently recrystallised or had their U-Pb ages reset during Paleoproterozoic orogenesis. If this were the case, the reported absence of Neoproterozoic detrital rutile in the Songshan Group could not be attributed to the absence of high-P Neoproterozoic metamorphic rocks in their source region.

The authors need to acknowledge these inherent caveats to interpreting rutile U-Pb ages in metamorphic rocks. They should then justify their interpretation that the detrital rutile grains in the Songshan Group are neocrystallised grains that grew during 2.0–1.7 Ga metamorphic event rather than being recrystallised or reset older detrital grains. I believe several observations already presented in the paper support the original interpretation made by the authors and should be more explicitly highlighted:

- Some older detrital rutile grains are preserved in the Songshan Group (Line 172). This suggests temperatures were not high enough during Paleoproterozoic greenschist facies metamorphism to reset the U-Pb system of rutile. Given that resetting of the U-Pb system is a function of grain size, this argument could be further supported by demonstrating that the older detrital grains in these samples are of a similar size to the dominant 2.0—1.7 Ga population. The authors report the size of rutile grains in the supplementary files, so quantifying the relationship between grain size and age should be simple.
- Zr-in-Rutile temperatures are low and uniform, which would not be expected if these were originally Neoproterozoic detrital grains derived from high-T metamorphic rocks elsewhere in the NCC. Zr diffusion in rutile is very slow and probably has a closure temperature on the order of ~700 C. This makes it unlikely that the Zr concentrations of rutile would have been re-equilibrated during greenschist facies metamorphism at 2.0—1.7 Ga. Thus, the authors can rule out that the abundant 2.0—1.7 Ga rutile grains in the Songshan Group represent originally ~2.5 Ga detrital grains that have had their U-Pb systematics reset during the 2.0—1.7 Ga event.
- The samples appear to be compositionally (super-)mature Qtz-rich rocks that are presumably Ca- and Fe-poor. Ca- and Fe-poor bulk compositions would not favour conversion of detrital rutile grains to ilmenite or titanite during prograde metamorphism. Thus, the Songshan Group has a favourable composition to preserve older detrital rutile grains, if they were present (and as above, the U-Pb data indicates that some older ages are indeed preserved).
- The authors note that rutile in Songshan Group are aligned with dominant foliation as evidence of metamorphic origin. I agree, but it would be nice to show a photomicrograph in the main body of the manuscript demonstrating this textural evidence.

Detrital zircon evidence

The authors suggest that only the Wufoshan Group contains detrital zircons derived from metamorphic sources, suggesting that only the Paleoproterozoic orogenic event produced a sufficient volume of thickened orogenic crust to be sampled by the sedimentary archive.

The only evidence supporting this interpretation that is presented in the main body of the paper is that ‘...only the Wufoshan Group contains zircons with numerous core-rim structures...’ (lines 286—287). However, from the CL images of detrital zircon grains from the Songshan Group presented in the supplementary material (Fig. S5), I can see several grains that apparently have secondary overgrowths or zones that truncate oscillatory-zoned core domains (e.g., grains 13, 9, 2 on Panel A, grains 14 and 16 on panel B, grain 3 on panel C, grains 17 and 32 on panel E, grain 2 and 26 on Panel F). The argument that

core-rim structures are more common in detrital zircons from the Wufoshan Group would be more compelling if the authors could quantify how common these textures are in both the Songshan and Wufoshan groups (i.e. what % of grains have core-rim structures in each group?).

The authors also cite Fig. S9 as containing additional evidence for abundant metamorphic-derived zircons in the Wufoshan Group (lines 286—287) but do not discuss the significance of the data presented in this supplementary figure. Fig S9 summarises REE patterns for zircons from the Wufoshan Group and shows that some grains have HREE-depleted signatures, which are typical of zircon that has co-crystallised with garnet, potentially indicating high-P metamorphism. However, these REE-patterns are not tied to the corresponding U-Pb age of the grains. Furthermore, the HREE-depleted measurements apparently come from the cores of zircon grains that are surrounded by metamorphic rims, which the authors appear link to the main Paleoproterozoic metamorphic event. Based on this data, I have two queries:

(1) How can the authors be certain these HREE-depleted metamorphic zircons aren't pre-2.0 Ga grains and hence actually provide evidence for high-P metamorphism related to the earlier (Neoproterozoic) orogenic event(s). I note from Fig. 6F that the Wufoshan Group contains numerous pre-2.0 Ga detrital grains. If the authors could demonstrate that all of the HREE-depleted zircons in the Wufoshan Group are ~1.7-2.0 Ga, it would strengthen the interpretation that the Paleoproterozoic metamorphic event was characterised by higher pressure metamorphism compared to earlier orogenic events.

(2) I am curious as to why similar detrital zircon REE data has not been presented from the Songshan Group. A comparison of time-constrained REE data from detrital zircons from both the Wufoshan and Songshan groups would make for a much more balanced comparison of the metamorphic style of Neoproterozoic vs. Paleoproterozoic orogenesis based on detrital zircon record (i.e., demonstrate that Neoproterozoic detrital zircon populations lack HREE-depleted signatures, whereas Paleoproterozoic zircon populations do contain them).

Minor comments (Line-by-line)

Line 1 (title): I would remove 'dramatic' from the title. Much of the evidence for a change in orogenic style in the NCC is from models and datasets beyond the foreland basin deposits that this paper focuses on. I think the evidence from the sedimentary record is more subtle but compelling nonetheless (as is always the case for reconstructing orogenic histories from sedimentary rocks).

Line 53—54: Somewhat redundant to describe an 'Alpine-style' fold-and-thrust belt associated with a 'Himalayan-style' collisional orogeny. Describing it as a fold-and-thrust belt developed during a Himalayan- (or Alpine) style collision would suffice.

Line 70: Replace 'loosely' with 'incompletely' documented.

Line 113: The clasts (not 'gravels') in the conglomerate are aligned with the tectonic foliation.

Lines 133: suggest rewording: The sandstones is composed of well-rounded detrital quartz grains that contain dusted rims and quartz overgrowths.

Line 216: Typo: 'provenances' should be provenance.

Line 217: Typo: 'filtration' should be filtering.

Line 276: '...could be linked to a secular change of orogenesis'. This clause isn't needed here. The point being made is that rutile forms in high-pressure metamorphic rocks. This is true regardless of when high-pressure rocks first appeared in the geological record (which is more relevant when this point is discussed later in the paper).

Line 278: Typo: 'event' should be events.

Line 282: Typo: 'metamorphic origins' should be metamorphic origin.

Line 288: 'indicate' should be replaced with '....are interpreted to reflect....' or 'record'.

Line 295: unclear what is meant by '....where the largest accretion and crustal growth occurred'. Largest accretion and growth compared to what? The local geology or the globally? Or do you mean Altaid-style orogens are the sites of significant crustal accretion and growth?

Line 299: final sentence of this paragraph is vague. Do you mean rutile and low-Lu and low-Lu/Dy zircons are rare at this time in the global detrital record? If you want to make this point, you need to explain the significance of low-Lu and low-Lu/Dy zircons.

Line 305: I'm not convinced that high-pressure granulites (and to some extent eclogites) provide evidence of 'cold subduction'. Granulites aren't common in 'cold' subduction zones and perhaps only Lawsonite eclogites provide compelling evidence for 'cold' subduction. If there are metamorphic rocks that do record low T/P thermal gradients in the NCC, these should be highlighted.

Line 317—319: As written it this sentence implies that only the tectonic events in the NCC influenced the wider Earth system. I think the authors mean that the changes recorded in the NCC are typical of those preserved in many early Paleoproterozoic terranes, suggesting a global link between a transition in tectonic regime and surficial environments. As such, it would be helpful to cite some other examples where similar Archean-Paleoproterozoic tectonic transitions are preserved in the structural, sedimentary, magmatic record (e.g., Western Australia, North America, South Africa...).

Line 319—320: Are you saying this style of late Archean cratonisation is typical of the NCC or all cratons? If the latter, need to support with references.

Line 325: Typo: 'Nutrition' should be nutrients. For those not familiar with biochemical cycles, it would be useful to elaborate on what these nutrients actually are and how they form an important component of orogenic rocks.

Line 334—336: Might be helpful to remind the readers that these are quartz-rich clastic rocks, which provides evidence that the more labile minerals in their source TTGs/greenstones were indeed destroyed during weathering.

Line 354—356: A citation to Cawood et al. 2022 *Reviews of Geophysics*, v. 60 (4) would be appropriate here.

Line 358: 'higher peak metamorphic pressures from minerals or increased high-pressure rock record'. This is vague, aren't these the same thing? i.e., high-pressure rocks are defined by having high-pressure minerals.

Line 362—367: Unclear if this is a general statement about all orogens or just the study area.

Line 368—377: The concluding paragraph linking the stabilisation of continents to the boring billion and the appearances of eukaryotes is essentially a throw-away line. In the context of this paper, the link between continental stabilisation, the boring billion, and eukaryote evolution is not clear. Is it to do with ocean chemistry? tectonics? What new insights on these topics does this paper make? A stronger finish would be to reinforce the sedimentary response to changing orogenic styles, which is really what this paper does a nice job of doing.

Line 379: (Methods). No data for reference materials is presented in the supplementary material. It is stated that 'all ages are consistent with recommended values' but the original data for the reference materials must be included so their veracity can be assessed independently.

General comments on grammar

- The definitive article (the) is used incorrectly in several places throughout the manuscript. I encourage the native English-speaking co-authors or journal type editors to revise.
- A more pedantic point, but the use of forward slashes (/) obscures the intended meaning of the text in several places. It's not clear if the forward slash means 'or', 'and', or is taking the place of a hyphen. For example, arc/micro-continent collision. Is this describing a collision between an arc and a microcontinent (arc-microcontinent collision), or is it indicating there were multiple collisions, some involving arcs and microcontinents, others involving only arcs or only microcontinents? I suggest replacing '/' with 'and' or with 'or' for clarity.

Point-by-point response to the reviewers' comments (Reply in BLUE)

We are thankful to the reviewers for the thorough reviews, constructive comments, and suggestions, which have been very helpful for us to improve the quality of the paper. We appreciate these comments and suggestions, and implemented careful revisions and addressed all concerns accordingly. The detailed revisions can be found in the “**Manuscript with changes tracked (in red)**”, and the **point-by-point response** as follows (reply in blue).

Reviewer #1 (Remarks to the Author):

During the late Archean to Paleoproterozoic, the Earth's deep and surficial reservoirs underwent dramatic changes, as evidenced by a number of indications, which are essential to comprehending the variation and interaction of Earth's solid and surface systems. Nevertheless, the transformative mechanism by which plate subduction and plate boundary evolved at the critical Archean to Paleoproterozoic juncture still remains enigmatic. This contribution reports the identification and life cycle of an ancient Paleoproterozoic foreland basin in the North China Craton, one of the oldest foreland successions in the world, and highlights the long-term evolution from ca. 2.5-2.47 Ga deposition initiated by warm subduction and Altaid-style “soft” micro-continental collision and deformation-uplift-denudation due to ca 1.97-1.85 Ga Himalayan-style continental collision. This study adds to our understanding of Earth's early evolution by shedding light on the multiple stages of source-and-sink history of the foreland sediments in response to late Archean to Paleoproterozoic orogenic episodes as well as potential influences on the dramatic variations of surficial processes and conditions on early Earth. Despite that, there are still some concerns with this contribution, prompting me to recommend a minor revision prior to publishing.

We appreciate Prof. Wenjiao Xiao for the positive recommendation and constructive comments. We have implemented relevant revisions and addressed all concerns accordingly.

Major concerns:

(1) Priority one is the connection between the tectonic evolution of the two episodes of orogenesis in North China Craton. Their respective tectonic evolution has been clearly manifested by the convincing works of the authors' group in recent years, however, the successive evolution history in the west of the “Proto-Eastern Block” (Figure 7A) is, in my

viewpoint, an important element for constructing the tectonic architecture from “supercraton” to “supercontinent” over a timeframe of ~500 million years (i.e., from ~2.5 Ga to ~1.98 Ga) (Figures 7A-B). In any case, an ancient oceanic basin lay between the Proto-Eastern Block and Ordos Block. If this ocean was consumed by subduction zone along the western margin of the Proto-Eastern Block, (i.e., the COB transferred to an active margin), is there any geological (magmatic, metamorphic, or accretionary) record for the successive prolonged subduction from the late Archean to Paleoproterozoic? Alternatively, if the COB was indeed a passive margin, were sediments equivalent to the passive boundary deposited and preserved? In other words, the tectonic scenario between two episodes of orogenesis in the Archean and Proterozoic is likely a crucial link for elucidating the transformative mechanism for the systematic changes from deep to surface in early Earth.

Thank you for this insightful comment and discussion.

Indeed, the long-term evolution of the North China Craton (NCC) is complex. Nevertheless, the two major tectono-magmatic-thermal events (orogenesis or cratonization) have been well identified during the past two decades. However, several different tectonic models have been proposed based on the emphasis of different evidence. The core debate is about the detailed evolutionary history of the early Paleoproterozoic from 2.45 Ga to 2.0 Ga. For example, Zhai et al. (2005, 2011) suggested that this period experienced post-cratonisation rifting-subduction-collision along three Paleoproterozoic mobile belts. Zhao et al. (2005, 2012) suggested that this period is characterized by prolonged subduction of a paleo-ocean since 2.56 Ga between the Eastern Block and Western Block, which was terminated by collisional orogenesis at ca. 1.85 Ga. Kusky et al. (2003, 2016) suggested that there are multiple stages of accretionary and collisional orogenic events from ca. 2.5 Ga to 1.8 Ga, which led to the outward growth of the NCC and final incorporation into the Columbia supercontinent. In these models, the Songshan sedimentary succession was interpreted as deposition along the extensional rift or cratonic margin that unconformably overlain the Archean basement (Zhai et al., 2005), retro-arc foreland basin (Zhao et al., 2012; Liu et al., 2012), or peripheral to retro-arc foreland basins (Kusky et al., 2016), respectively. The latter two models are very similar regarding the tectonic affinity of the succession, which is related to the eastward subduction of oceanic lithosphere beneath the (Proto-)Eastern Block.

Based on our new datasets and compilation of regional magmatic, metamorphic, and accretionary records, we can constrain the Neoproterozoic arc-(micro)continent collision event that

terminated the westward subduction of the oceanic basin between the central arc system and the Eastern NCC (we termed the Proto-Eastern Block). The arc-microcontinent collision may have been followed by a subduction polarity reversal event (e.g., Kusky, 2011; Wang et al., 2015; Kusky et al., 2016), leading to the eastward subduction beneath the western margin of the Proto-Eastern Block, which converted as an Andean-type active continental margin. The lowest section of the Songshan foreland is in response to the arc–(micro)continent collision (Fig. 8a), while the upper section may be a response to the long-term retro-arc sedimentation. The closure processes of the oceanic basin between the Proto-Eastern Block and Ordos Block remain poorly constrained due to the limited outcrop of the Ordos Block, but a recent publication suggests a complex subduction-accretion process based on a summary of new regional data sets (Yin et al., 2023, ESR). During the Paleoproterozoic orogenesis, the previous Songshan foreland successions were converted into a fold-and-thrust belt (Fig. 8b). Our data provide a better constraint on these two stages of orogenesis and their distinctive styles (e.g., Altaid-style vs. Himalayan-style).

We have revised the relevant paragraphs to clarify the 2.5–2.0 Ga evolution in the main text (see Lines 284–289), along with a detailed summary of the geological background and available tectonic models of the NCC in the Supplementary Note.

- Zhai, M., Guo, J. & Liu, W. Neoproterozoic to Paleoproterozoic continental evolution and tectonic history of the North China Craton: a review. *J. Asian Earth Sci.* **24**, 547-561 (2005).
- Zhai, M. Cratonization and the Ancient North China Continent: A summary and review. *Science China Earth Sciences.* **54**, 1110-1120 (2011).
- Zhao, G., Sun, M., Wilde, S. A. & Li, S. Late Archean to Paleoproterozoic evolution of the North China Craton: key issues revisited. *Precambrian Res.* **136**, 177-202 (2005).
- Zhao, G. et al. Amalgamation of the North China Craton: Key issues and discussion. *Precambrian Res.* **222-223**, 55-76 (2012).
- Kusky, T. M. & Li, J. Paleoproterozoic tectonic evolution of the North China Craton. *J. Asian Earth Sci.* **22**, 383-397 (2003).
- Kusky, T. M. et al. Insights into the tectonic evolution of the North China Craton through comparative tectonic analysis: A record of outward growth of Precambrian continents. *Earth-Sci. Rev.* **162**, 387-432 (2016).
- Liu, C. et al. Detrital zircon U–Pb dating, Hf isotopes and whole-rock geochemistry from the Songshan Group in the Dengfeng Complex: Constraints on the tectonic evolution of the Trans-North China Orogen. *Precambrian Res.* **192-195**, 1-15 (2012).
- Kusky, T. M. Geophysical and geological tests of tectonic models of the North China Craton. *Gondwana Res.* **20**, 26-35 (2011).

Wang, J., Kusky, T., Wang, L., Polat, A. & Deng, H. A Neoproterozoic subduction polarity reversal event in the North China Craton. *Lithos.* **220-223**, 133-146 (2015).

Yin, C. et al. Paleoproterozoic accretion and assembly of the Western Block of North China: A new model. *Earth-Sci. Rev.*, 104448 (2023).

(2) The structural style of the foreland deposits matches with the characteristics of the thrust-fold belts at the middle-to-upper crustal-level. Although rutiles have been utilized to constrain the metamorphic temperature (T) for their presumed sources, it is necessary to directly quantify the metamorphic T for the schists in the thrust-fold belts and to, on the other hand, mutually verify the provenance of the detrital rutiles.

Thank you for highlighting this. We have supplemented the P - T conditions of representative samples from the quartz-mica schists based on Ti-in-biotite geothermometer and phase equilibria modelling (Supplementary Note, Figure S11, Table S3). Our results confirm that the majority of rutiles in the Songshan Group are of metamorphic origin formed during the Paleoproterozoic collisional event. The Zr-in-rutile temperature is comparable to the Ti-in-biotite temperature of the schist.

(3) There must be some clerical errors in the text depicting the age peaks for the geochronologic spectra (Figure 6). For instance in line 173, one of the age peaks for detrital rutile is written as “1.87 Ga” but is shown as “1.86 Ga” in Figure 6E; in line 247, age peaks for detrital zircons are written as “1.93 Ga and 1.84 Ga” but are shown as “1.94 Ga and 1.83 Ga” in Figure 6G; etc. Please check thoroughly the consistency of the figure and text for the zircon and rutile U-Pb geochronology.

Thank you for this important point. We carefully checked the figures and corresponding descriptions in the main text, which now are consistent between the main text and figures.

A few specific comments:

(1) Lines 34-35, change “one of most...” to “one of the most...”;
Corrected.

(2) Line 246, change “(Fig. 6G)” to “(Fig. 6F)”;
Corrected.

(3) Figures 3A and 3B should have photographic scale.

Thanks. We added the scale.

Wenjiao Xiao

We deeply appreciate Prof. Wenjiao Xiao for the above constructive comments.

=====

Reviewer #2 (Remarks to the Author):

After carefully reviewing this submission, I don't think it can be published in NC, I recommend reject, the main reasons for my suggestion are as follows.

We appreciate Reviewer #2 for the critical comments and suggestions, most of which have been essential for us to improve the quality of the manuscript. **Nonetheless, it appears that there might be a slight misapprehension regarding the core focus of our paper.**

Our study focuses on elucidating the styles of two stages of orogenesis in the Neoproterozoic and Paleoproterozoic and their possible impacts on the surficial environment. This multifaceted understanding is derived from integrated structural, petrological, and geochronological investigations on one of the most ancient foreland basins in the North China Craton (NCC), along with big-data comparison. It is important to clarify that our intention extends beyond the singular aim of deciphering the regional tectonic evolution of the NCC, as mentioned by the reviewer.

1. It seems that this study is based on a relatively small number of samples - especially given that you are trying to resolve large scale issues relating to the temporal development of the Paleoproterozoic foreland successions. In some cases, I found it difficult to actually determine how many samples had been used. It is imperative that you provide a clear statement regarding the exact number of samples used for each type of analysis - it is absolutely critical that it you can demonstrate that you have used sufficient samples to justify the conclusions that you have reached.

Our data sets include two aspects, including **new analyses** (745 zircon/rutile U-Pb isotopic analyses, 90 trace elemental analyses of rutile, 4 Ar-Ar dates, 171 mineral chemical analyses) and **literature compilation** as we stated in the main text (including in the previous version) (**Supplementary Data 1–4**). **The numbers of detrital zircon ages and rutile ages used in the discussion are 1365 and 136, respectively**, which cover the main units of the Songshan and Wufoshan groups and are sufficient to make a convincing analysis of their sedimentary-to-deformational evolution. Furthermore, we conducted comparisons with global rutile and detrital zircon datasets (**Figs. 1a, 6, 7g**) and **have revised relevant figures (Supplementary Figs. S6, S7) and text to enhance clarity**.

We have labelled the analysis numbers of zircons and rutiles in each sample in **Fig. S6a (now Fig. S7a)** in the previously submitted version. To make it clearer, we also added the analysed numbers of every sample in **Fig. S6** in this revised version.

2. This study has no novelty and too local, the methods are simple, many publications had defined the two-stages orogenesis model of the North China Craton, Archean mélangé and Paleoproterozoic foreland successions developed within the Trans-North China Orogen (see Professor M. Santosh's group papers, Professor Guo-chun Zhao's group papers, Professor Ming-guo Zhai's group papers, Professor T. Kusky's group papers, etc.)

Thanks for this critical comment, but the reviewer may have misunderstood the innovation points of this manuscript.

As we have responded, the tectonics of the NCC has been a hot topic, with one of most important progresses being the documentation of two stages of tectono-magmatic-thermal events in the Neoproterozoic (ca. 2.5 Ga) and Paleoproterozoic (ca. 1.9 Ga) (attributed to two-episode orogenesis by some) in the past two decades (references mentioned by Reviewer #2). However, the nature and detailed process of these two stages of events (orogenesis) remain debated, with several models being proposed to explain the regional tectonic evolution of the NCC (**Supplementary Note**). Thus, this is not to say all tectonic models on regional tectonics are perfect and no further research on regional tectonics and more general scientific questions is needed.

Rather, the in-depth discussion and constraints on some general scientific questions in Earth Science regarding the styles of orogenesis and plate tectonics and their environmental effects remain limited and further investigations, which have been the focus

and frontier of recent studies (e.g., Zhai and Peng, 2020; Cawood et al., 2022). The NCC, with its two stages of ‘orogenic’ events, offers a unique natural laboratory for investigating these exciting scientific inquiries.

In this contribution, we focus on such a hot research topic from unique sedimentary and structural perspectives of the orogenic foreland, one of the oldest foreland basins from the NCC. Our integrated field, structural, petrological, and geochronological data sets allow us to better constrain the distinctive orogenic styles in the Neoproterozoic and early Paleoproterozoic. Furthermore, we establish connections between orogenesis and surface environmental changes, yielding fresh insights into the evolution of orogenic styles and the interactions between solid Earth processes and near-surface systems.

We believe that Reviewer #2 might be aware of this research trend and will understand the novel aspects and motivations of our study as we explained in Introduction and Discussion in detail.

Zhai, M. & Peng, P. Origin of early continents and beginning of plate tectonics. *Sci. Bull.* **65**, 970-973 (2020).

Cawood, P. A. et al. Secular Evolution of Continents and the Earth System. *Rev. Geophys.* **60**, (2022).

3. Many concepts and descriptions are confused in the manuscript, for example, Nuna is different to Columbia supercontinent, see JG Meert et al. 2012 Gondwana Research; JG Meert and M Santosh 2017 Gondwana Research. Also, many Chinese strata names confused me, didn't explain them clearly. The citations are not rigorous in the manuscript, who was the first to propose this concept of Columbia supercontinent, the evolution models of Columbia supercontinent and North China Craton??? The authors even didn't add any references in many paragraphs/sentences, see lines 323, 360-367, etc.

Thanks for this critical comment. We reply to these comments one by one.

(1) Terminology of Nuna vs. Columbia:

The debate about the terminology of Paleoproterozoic supercontinent is not the main focus of this manuscript, but we simply explain why we used the ‘Columbia/Nuna’ in the first version. The Paleoproterozoic supercontinent in the past two-decade literature has been variably termed, such as ‘Nuna’ (Hoffman 1997), and ‘Columbia’ (Rogers and Santosh, 2002). The Nuna initially

refers to the connected continent of five large cratons including Baltica-North America-Siberia-Northern and Western Australia, which have been regarded as the core of the Paleoproterozoic supercontinent in later studies. This is the basis for reconstructing the Paleoproterozoic supercontinent. With time going on, more cratons have been linked and posited around the Nuna (e.g., Evans, 2003, *Tectonophysics*; Campbell and Allen, 2008, *Nature Geoscience*; Evans and Mitchell, 2011, *Geology*; Mitchell et al, 2012, *Nature*, 2021 *Nature Reviews Earth and Environment*; Wan et al., 2015, *Nature Communications*). The original proposal of ‘Columbia’ is largely based on evidence of rifts in India and North America (Columbia River) (Rogers and Santosh, 2002), however, this link seems to have not been widely used in later studies.

Thus, both Nuna and Columbia have been widely used to refer to the Paleoproterozoic supercontinent in the last decades (see a historical review by Meert, 2012). Currently, most researchers prefer to use the terms ‘Columbia (Nuna)’, ‘Nuna (Columbia)’, ‘Nuna/Columbia’ and even ‘Columbia-Nuna’ to avoid the debate about terms as we did before (e.g., Ernst et al., 2008; Li et al., 2023, *Earth-Science Reviews*; Wu et al., 2022, *G-cubed*). A recent review by Ross Mitchell et al. (2021) may provide a possible solution to this issue, i.e. Nuna as a mega-continent (like Gondwana) in the core of supercontinent Columbia (although the configuration is still debated).

Respectfully, we revised our terminology to ‘Columbia’ in the whole manuscript to avoid ambiguity following the reviewers’ suggestion, except for in the first appearance, where we wrote ‘Columbia, with the core of Nuna’.

(2) References about the supercontinent:

Considering the limitation of the number of references, we just cited the key and indispensable references in the main text and put some in the Supplementary Information in the first version. According to the reviewer’s suggestion, we added some key references about the NCC and its link with the Paleoproterozoic supercontinent in relevant places in the revised version (e.g., Zhao et al., 2002; Kusky et al., 2007; Wu et al., 2018, 2022).

Line 323: We add references here.

Lines 360-367: We add references here.

- Hoffman, P.F., Tectonic genealogy of North America. In Earth structure. In: van der Pluijm, B.A., Marshak, S. (Eds.), *An Introduction to Structural Geology and Tectonics*. McGraw-Hill, New York, pp. 459–464 (1997).
- Rogers, J. J. W. & Santosh, M. Configuration of Columbia, a Mesoproterozoic Supercontinent. *Gondwana Res.* **5**, 5-22 (2002).
- Evans, D.A. True polar wander and supercontinents. *Tectonophysics*, **362**(1-4), pp.303-320 (2003).
- Evans, D. A. D. & Mitchell, R. N. Assembly and breakup of the core of Paleoproterozoic-Mesoproterozoic supercontinent Nuna. *Geology*. **39**, 443-446 (2011).
- Campbell, I.H. and Allen, C.M. Formation of supercontinents linked to increases in atmospheric oxygen. *Nature Geoscience*, **1**(8), pp.554-558 (2008).
- Meert, J. G. What's in a name? The Columbia (Paleopangaea/Nuna) supercontinent. *Gondwana Res.* **21**, 987-993 (2012).
- Mitchell, R. N., Kilian, T. M. & Evans, D. A. D. Supercontinent cycles and the calculation of absolute palaeolongitude in deep time. *Nature*. **482**, 208-211 (2012).
- Mitchell, R. N. et al. The supercontinent cycle. *Nat. Rev. Earth Environ.* **2**, 358-374 (2021).
- Wan, B., Windley, B. F., Xiao, W., Feng, J. & Zhang, J. E. Paleoproterozoic high-pressure metamorphism in the northern North China Craton and implications for the Nuna supercontinent. *Nat. Commun.* **6**, 8344 (2015).
- Li, Z.X., Liu, Y. and Ernst, R. A dynamic 2000–540 Ma earth history: from cratonic amalgamation to the age of supercontinent cycle. *Earth-Science Reviews*.104336 (2023).
- Wu, C. et al. Paleoproterozoic Plate Tectonics Recorded in the Northern Margin Orogen, North China Craton. *Geochemistry, Geophysics, Geosystems*. **23**, e2022GC010662 (2022).
- Wu, C. et al. A 1.9-Ga Mélange Along the Northern Margin of the North China Craton: Implications for the Assembly of Columbia Supercontinent. *Tectonics*. **37**, 3610-3646 (2018).
- Zhao, G., Cawood, P. A., Wilde, S. A. & Sun, M. Review of global 2.1–1.8 Ga orogens: implications for a pre-Rodinia supercontinent. *Earth-Sci. Rev.* **59**, 125-162 (2002).
- Kusky, T., Li, J. & Santosh, M. The Paleoproterozoic North Hebei Orogen: North China craton's collisional suture with the Columbia supercontinent. *Gondwana Res.* **12**, 4-28 (2007).

4. The manuscript mixed use of full name and abbreviations here and there. For example, North China Craton and NCC; Circa and Ca.; Formation and Fm. (See Lines 52, 110, 123, 127-128, 140, 158, 201, 206, 317, 344, 661, 631).

Thanks for pointing out this. We have carefully checked this issue. Partially agree.

(1) North China Craton and NCC: We now use the full name in the whole manuscript.

(2) When starting a sentence, full spelling (e.g., circa) is required. So ‘Circa’ is correct in Line 344.

(3) For the abbreviation of Formation, 'Fm.' or 'Fm' are widely used. To make it consistent, we use the full name 'Formation' in the main text of the whole manuscript but use Fm in figures (to save space).

Line 123: 'the protoliths of the formation' is correct. 'the formation' refers particularly to the 'Wuzhiling Formation'.

Line 127-128: Like above. This is correct as well.

Line 140: We revise 'Formations' to 'formations'.

Line 158: This is correct.

Line 201 & 206: We use the full name 'Formation' here.

5. The logic of this manuscript is weak, this work can't better define the differences and controversies among the models. The maximum depositional ages of the detrital zircons can't be represented the depositional age of the strata, not to mention such old metamorphic strata; The age analysis lacks evidence, you can't say pb-loss casually. The authors can't discuss more contributions about the controversies among the models of lines 260-267 and can't get this conclusion of lines 288-290 without the evidence.

Thanks for the criticism. We address these concerns one by one.

(1) Tectonic models. As mentioned before, the reviewer should be aware that our main purpose is not to resolve the debate on the tectonic models regarding the long-term history of the NCC from Archean to Paleoproterozoic **but to provide sedimentary and structural constraints on the detailed sedimentary-to-deformation evolution of the Songshan foreland and better constrain the distinctive styles of two stages of orogenesis in the Neoproterozoic and Paleoproterozoic**. These two stages of orogenesis have not reached a consensus yet, and the styles of orogenesis also require further studies in detail. For example, in some models (e.g., Zhao et al., 2005, 2012), there are no detailed constraints on Archean collisional orogenesis. Our data provide new sedimentary evidence for the arc-microcontinent collision and terrane assembly in the late Archean to the earliest Paleoproterozoic. Our new data of rutiles and detrital zircons and structural deformation further show the different orogenic styles during the late Archean-early Paleoproterozoic (ca. 2.5 Ga) and middle Paleoproterozoic (ca. 1.9 Ga). **The entire evolution of the NCC from the Neoproterozoic to the Paleoproterozoic is out of the scope of this research and requires further investigation.**

- (2) Maximum depositional ages. Due to the lack of syn-depositional volcanic interlayers, the detrital zircons have been widely used to constrain the maximum depositional ages (MDA) of strata ranging from Archean to Phanerozoic in age (e.g., Zi et al., 2022, GCA). For Precambrian zircons, the Pb-loss has a significant influence on the MDA. We have carefully examined the age data and conducted data filtering, and we use conservative methods to estimate the MDA.
- (3) Age analysis and Pb loss. We filtered discordant data to preclude the influence of possible Pb-loss. This is important for Precambrian detrital zircon studies.

Zhao, G., Sun, M., Wilde, S. A. & Li, S. Late Archean to Paleoproterozoic evolution of the North China Craton: key issues revisited. *Precambrian Res.* **136**, 177-202 (2005).

Zhao, G. et al. Amalgamation of the North China Craton: Key issues and discussion. *Precambrian Res.* **222-223**, 55-76 (2012).

Zi, J., Rasmussen, B., Muhling, J. R. & Fletcher, I. R. In situ U-Pb and geochemical evidence for ancient Pb-loss during hydrothermal alteration producing apparent young concordant zircon dates in older tuffs. *Geochim. Cosmochim. Acta.* **320**, 324-338 (2022).

6. I found the conclusions to be somewhat weak - they really do not stress the main contributions of the paper. The Conclusions need to be improved.

We added a concluding paragraph to highlight and summarize our contribution from this study.

“In conclusion, this study provides new constraints on the depositional-to-deformational evolution of the Paleoproterozoic Songshan foreland — one of the oldest orogenic forelands in the North China Craton, from sedimentary and deformational perspectives. Importantly, it underscores the significance of multi-mineral geochronological analysis of key minerals from sedimentary archives in ancient orogenic forelands, which provide clues as to how sedimentation and metamorphism responded to changing orogenic style. Our research has shed new light on changes in tectonic processes during the Archean-Proterozoic transition and their influence on global biogeochemical cycles, providing an innovative method for investigating the evolution of foreland sedimentary basins. Additionally, our findings offer fresh perspectives on the dynamic interplay between solid and surficial Earth processes during the Paleoproterozoic, contributing to a deeper understanding of our planet’s evolution towards a more habitable environment.”

7. Comments on the figures:

Figure 1. Nuna or Columbia??? The timing of supercontinents are right?

As replied before, we have addressed this problem about the ‘Nuna’ or ‘Columbia’ (see our detailed response to question 3). **In the revised version, we use ‘Columbia’ in the whole manuscript** (except for the first appearance) to avoid any debate on the name of the Paleoproterozoic supercontinent. The timing and configuration of supercontinents (1.8 Ga) are based on **Wan et al., 2020, Science Advances** and **Wan et al., 2015, Nature Communications**.

Wan, B., Windley, B. F., Xiao, W., Feng, J. & Zhang, J. E. Paleoproterozoic high-pressure metamorphism in the northern North China Craton and implications for the Nuna supercontinent. *Nat. Commun.* **6**, 8344 (2015).

Wan, B. et al. Seismological evidence for the earliest global subduction network at 2 Ga ago. *Sci. Adv.* **6**, eabc5491 (2020).

Figures 2-5. the authors discussed the structures, not even providing the nature of faults in the Figure 2; the field interpretations seem unclear in the Figures 3-5; the authors should define the abbreviations in the figures. No any field data on the foliations and lineations in the Figure 3. Confused the COB and TNCO.

Thanks for pointing out this. We improved the figures accordingly.

(1) In Fig. 2, we provided the nature of faults.

(2) We added more structural data (foliation, lineation, fold) in Figures 3–5 (now **Figs 3, 4, S5**).

(3) All abbreviations have been clarified in the figure captions.

(4) The COB is used to describe the 2.5–2.47 Ga orogenic belt emphasized by Kusky et al., whereas the TNCO is used to describe the 1.95–1.8 Ga collisional orogen proposed by Zhao et al. They are not in conflict as they represent orogens formed or modified in different periods.

Figure 6. How does the author distinguish between the detrital/metamorphic ages?

Good point. Rutile grains from the unmetamorphosed Wufoshan Group are of detrital origin, while rutile grains from the Songshan Group are dominated by a metamorphic origin (age clustering at 2.0–1.8 Ga, n=62), with minor survived detrital rutiles (e.g., 2.50 Ga, n=3) (**Table S2**). The ca. 2.0–1.8 Ga metamorphic rutile in the Songshan Group has distinctive textures and

Zr-in-rutile temperatures compared with detrital rutile from the Wufoshan Group (see the new Fig. 7a–f). We also provided a discussion on this to make it clearer.

“The timing of metamorphism of the Songshan Group can be constrained by metamorphic rutile. The young rutile grains (<2.0 Ga) from the Songshan Group are interpreted to be of metamorphic origin based on the following lines of evidence: 1) the rutile grains exhibit a consistent alignment in the dominant foliation, and have euhedral, subhedral and irregular shapes (Fig. 7a–c); 2) the peak ages of 1.95 Ga and 1.87 Ga are markedly younger than those of detrital zircons (Fig. 5b–e); 3) the ZIR temperature of <2.0 Ga rutile grains is low (~469–586 °C) and comparable with the Ti-in-biotite temperature (~558–598 °C at 4 kbar) (Supplementary Tables 2–3); and 4) subordinate rutile grains with ages exceeding 2.0 Ga are inferred to have survived during prolonged Paleoproterozoic (ca. 2.0–1.8 Ga) metamorphism, implying that the peak metamorphic temperature of the Songshan Group was lower than the closure temperature (~600 °C) of rutile in the U–Pb isotope system.”

Figure 7. Too general and it is difficult to compelling to the readers because many places lack evidence, especially the time limits; also confused the COB and TNCO. It seems that the authors just want to mix everything together and doesn't clarify geological history very well. This manuscript does not contribute in terms of time constraints, it is too broad.

This figure (now Fig. 8) just aims to show the two types of orogenic styles based on our new sedimentary, structural, and geochronological evidence, and regional geological data, rather than providing a comprehensive history from Neoproterozoic to Paleoproterozoic. The detailed evolution of the whole NCC from 2.5–2.0 Ga is not the main scope of the paper, and it requires further detailed studies of the magmatic, sedimentary, and metamorphic records across the craton, which cannot be made in a relatively concise paper.

Overall, we appreciate the critical comments from reviewer #2, which pushed us to better present the work and clarify the novelty of this research.

Reviewer #3 (Remarks to the Author):

Huang and colleagues present a detailed study of the structural history, basin evolution, and provenance of Paleoproterozoic sedimentary rocks in the North China Craton. They integrate these datasets to infer a contrast in the tectonic style of orogenesis and continental assembly in the North China Craton (NCC) between the late Neoproterozoic and early Paleoproterozoic. The paper concludes by discussing the possible influences the purported transition in global tectonic regimes across the Archean-Proterozoic may have had on the wider Earth system.

Linking solid Earth processes and tectonic regimes to the wider Earth system is a highly active area of research and of interest to a broad cross section of the scientific community. Multi-disciplinary studies such the one presented in this paper are essential to developing a holistic understanding of relationship between tectonic processes and surface environments through deep time. The paper by Huang and colleagues takes the novel approach of studying the history of foreland sedimentary basins to document changes in tectonic regimes across the Archean-Proterozoic transition and its impact on global geochemical cycles. The paper is mostly well-written and is exceptionally and informatively illustrated. In my opinion, this paper would make an excellent contribution to Nature Communications. However, I have a number of queries regarding the provenance data that should be addressed before publication.

We deeply appreciate Reviewer #3 for the positive recommendation and very detailed comments regarding the data presentation and discussion, which are impressive and constructive. We have implemented relevant revisions and addressed all concerns accordingly.

Detrital Rutile evidence

One of the main pieces of evidence cited for a change in orogenic style from hot ‘soft’ collision in the Neoproterozoic to hard ‘cold’ collision in the Paleoproterozoic is the near absence of Neoproterozoic detrital rutile in the Songshan Group contrasting with the abundance of Paleoproterozoic detrital rutile grains in the Wufoshan Group. This interpretation is consistent with the tendency of rutile to grow in high-P metamorphic rocks, which are more likely to be exhumed and eroded into adjacent sedimentary basins in ‘colder’ orogens. Some important caveats to this interpretation are the tendency of rutile to recrystallise to ilmenite or titanite during the greenschist- to amphibole facies of a prograde metamorphic path (e.g., Ashley and

Law, 2015, contrib. Mineral. Petrol., 169) and the relatively low closure temperature of Pb diffusion in rutile (~500-600 C = greenschist-amphibolite facies conditions).

The fact that the Songshan Group has been metamorphosed to greenschist facies raises the possibility that Neoproterozoic detrital rutile grains were originally present in these sedimentary rocks but were subsequently recrystallised or had their U-Pb ages reset during Paleoproterozoic orogenesis. If this were the case, the reported absence of Neoproterozoic detrital rutile in the Songshan Group could not be attributed to the absence of high-P Neoproterozoic metamorphic rocks in their source region.

The authors need to acknowledge these inherent caveats to interpreting rutile U-Pb ages in metamorphic rocks. They should then justify their interpretation that the detrital rutile grains in the Songshan Group are neocrystallised grains that grew during 2.0—1.7 Ga metamorphic event rather than being recrystallised or reset older detrital grains. I believe several observations already presented in the paper support the original interpretation made by the authors and should be more explicitly highlighted:

Thank you very much for the constructive comments and solution ideas. We agree that there is the possibility that the Neoproterozoic detrital rutile grains were originally present in the sedimentary rocks and then experienced recrystallization or had their U-Pb ages reset during Paleoproterozoic orogenesis. As mentioned by reviewers, several lines of evidence already suggest that this may be not the case for the rutile in the Songshan Group. The rutile texture, age, Zr-in-rutile temperature, and existence of > 2.0 Ga rutile grains together imply the older rutile has not been reset by ca. 2.0–1.8 Ga metamorphism. The 2.50 Ga detrital rutiles (n=3) in the Songshan Group suggest relatively less high-pressure metamorphism in the Neoproterozoic. We add more evidence and discussion according to your suggestions to support our original interpretations, with revisions in **Lines 222–229 and Lines 313–332** and one-by-one replies for the later comments.

- Some older detrital rutile grains are preserved in the Songshan Group (Line 172). This suggests temperatures were not high enough during Paleoproterozoic greenschist facies metamorphism to reset the U-Pb system of rutile. Given that resetting of the U-Pb system is a function of grain size, this argument could be further supported by demonstrating that the older detrital grains in these samples are of a similar size to the dominant 2.0—1.7 Ga population. The authors report

the size of rutile grains in the supplementary files, so quantifying the relationship between grain size and age should be simple.

This is a good point. The closure temperature of rutile is ~ 600 °C for rutile grains of ~ 100 μm (Cherniak, 2000, CMP; Pereira and Storey, 2023). The sizes of older rutile grains (~ 60 – 130 μm) are similar to those grains (~ 55 – 130 μm , except for one < 50 μm) with ages of 2.0–1.7 Ga from the Songshan Group (see figure below). The few older rutile grains recorded in the Songshan Group suggest that the temperature of Paleoproterozoic (ca. 2.0–1.8 Ga) metamorphism is lower than 600 °C of the U-Pb system of rutile. Additionally, the pressure-independent Zr-in-rutile geothermometer results (469–586 °C) and Ti-in-mica geothermometer of the schist in the lower Songshan Group also support that the metamorphic temperature is not high enough to reset the rutile U-Pb system. Phase equilibrium modelling of a quartz mica schist also suggests rutile can be stable over a large P–T condition (Supplementary Fig. S11). **Collectively, we suggest that the ca. 2.0–1.8 Ga metamorphism has not reset the U–Pb age of possible older rutile in the Songshan Group.**

Length vs. Age of rutile in the Songshan Group. The size of older ones is within the range of ca. 2.0–1.8 Ga metamorphic rutile.

Cherniak, D. J. Pb diffusion in rutile. *Contrib. Mineral. Petrol.* **139**, 198-207 (2000).

Pereira, I. & Storey, C. D. Detrital rutile: Records of the deep crust, ores and fluids. *Lithos.* **438-439**, 107010 (2023).

- Zr-in-Rutile temperatures are low and uniform, which would not be expected if these were originally Neoproterozoic detrital grains derived from high-T metamorphic rocks elsewhere in the NCC. Zr diffusion in rutile is very slow and probably has a closure temperature on the order of

~700 C. This makes it is unlikely that the Zr concentrations of rutile would have been re-equilibrated during greenschist facies metamorphism at 2.0—1.7 Ga. Thus, the authors can rule out that the abundant 2.0—1.7 Ga rutile grains in the Songshan Group represent originally ~2.5 Ga detrital grains that have had their U-Pb systematics reset during the 2.0—1.7 Ga event.

Good point, Zr-in-rutile temperatures of rutiles from the Songshan Group are relatively low and uniform (Fig. 7e, Table S2), and the estimated metamorphic temperatures from quartz-mica schists by Ti-in-biotite geothermometer are about 558–598 °C (at 4 kbar) (Supplementary Note 4, and Table S3), which do not exceed the closure temperature 700 °C of Zr diffusion in rutile.

New Fig. 7e–f shows Zr-in-rutile temperatures of the Songshan and Wufoshan groups. The Songshan rutile is mostly of metamorphic origin (<2.0 Ga) and records lower uniform temperatures. Two ca. 2.5 Ga detrital rutile grains have higher temperatures (~657 °C).

- The samples appear to be compositionally (super-)mature Qtz-rich rocks that are presumably Ca- and Fe-poor. Ca- and Fe-poor bulk compositions would not favour conversion of detrital rutile grains to ilmenite or titanite during prograde metamorphism. Thus, the Songshan Group has a favourable composition to preserve older detrital rutile grains, if they were present (and as above, the U-Pb data indicates that some older ages are indeed preserved).

This is true, the quartzite and quartz-mica schist samples from the Songshan Group are compositionally poor in calcium and iron. The quartzite consists mainly of quartz and minor mica, and the quartz-mica schist consists of mica and quartz, with minor feldspar, magnetite, zircon and rutile (Fig. S3). We carefully checked the petrographic characteristics of the samples from the Songshan Group under SEM and EDS and didn't observe ilmenite or titanite in these samples. Thus, we suggest that the few older rutile grains have not been transformed into ilmenite or titanite during prograde or retrograde metamorphism.

- The authors note that rutile in Songshan Group are aligned with dominant foliation as evidence of metamorphic origin. I agree, but it would be nice to show a photomicrograph in the main body of the manuscript demonstrating this textural evidence.

New Fig. 7a–d shows the *in-situ* rutile textures and different shapes between the Songshan and Wufoshan groups.

Good suggestion. We added several photomicrographs (Fig. 7a–d) to show the textural evidence that supports rutile grains are of metamorphic origin in the main text. Therefore, based on the above evidence, we interpret that the majority of rutile grains in the Songshan Group are neocrystallised grains during 2.0–1.7 Ga. The striking difference between the scarcity of Neoproterozoic detrital rutile in the Songshan Group and the abundance of Paleoproterozoic detrital rutile grains in the Wufoshan Group can support the inference that a change in metamorphic and orogenic styles from the Neoproterozoic to the Paleoproterozoic.

Detrital zircon evidence

The authors suggest that only the Wufoshan Group contains detrital zircons derived from metamorphic sources, suggesting that only the Paleoproterozoic orogenic event produced a

sufficient volume of thickened orogenic crust to be sampled by the sedimentary archive. The only evidence supporting this interpretation that is presented in the main body of the paper is that ‘...only the Wufoshan Group contains zircons with numerous core-rim structures...’ (lines 286—287). However, from the CL images of detrital zircon grains from the Songshan Group presented in the supplementary material (Fig. S5), I can see several grains that apparently have secondary overgrowths or zones that truncate oscillatory-zoned core domains (e.g., grains 13, 9, 2 on Panel A, grains 14 and 16 on panel B, grain 3 on panel C, grains 17 and 32 on panel E, grain 2 and 26 on Panel F). The argument that core-rim structures are more common in detrital zircons from the Wufoshan Group would be more compelling if the authors could quantify how common these textures are in both the Songshan and Wufoshan groups (i.e. what % of grains have core-rim structures in each group?).

Thank you for the insightful comment and suggestion.

Yes, most zircon grains from the samples in the lower Songshan Group have textures of magmatic origin, with minor zircons having a very thin rim, which might be metamorphic overgrowth (either detrital metamorphic rim or post-depositional recrystallization). However, the percentage is relatively low, and the rim is too thin to be dated (Supplementary Fig. 6a–f).

We counted all zircon grains (randomly mounted) and evaluated the proportions of grains with core–rim textures in every sample. The statistical results suggest that the percentage of zircons with **core–rim structures from the Songshan group is ~6%, and the percentage of zircons with core–rim structures from the Wufoshan group is ~21%** (this didn’t include homogenous metamorphic zircons without rim).

In addition, most zircons from the Songshan Group show oscillatory zoning as described in the Supplementary Note and shown in Fig. S6, suggesting that they are mainly derived from magmatic sources. Therefore, we suggest that the Wufoshan Group contains detrital zircons from a metamorphic source.

We provided detailed information on different samples as follows.

- **Sample from the Wufoshan Group (~21% zircons with metamorphic rims):**
Sample 20DF03: total zircons (n=212), grains with core-rim structures (n=45), **~21%**;
Note that metamorphic zircons with homogenous banded or broad zoning (e.g., some

of them with HREE-depleted characteristics, Fig S8a-b) was not counted here. The percentage of overall metamorphic zircon should be higher than 21%.

- **Samples from the Songshan Group (~5.6% zircons with thin bright rims):**

Sample 21SS25: total zircons (n=224), grains with core-rim structures (n=0), 0%;

Sample 20SS05: total zircons (n=220), grains with core-rim structures (n=9), ~4%;

Sample 20SS06: total zircons (n=212), grains with core-rim structures (n=7), ~3%;

Sample 20SS01: total zircons (n=211), grains with core-rim structures (n=20), ~9%;

Sample 20SS07: total zircons (n=212), grains with core-rim structures (n=17), ~8%;

Sample 20SS09: total zircons (n=121), grains with core-rim structures (n=14), 12%;

Collectively, 67 grains have thin rims among 1200 (randomly mounted) counted zircon grains = 5.57% (typical zircons with rims are shown in Fig. S6a-f).

We incorporated relevant information into Supplementary Note 2.1.1 when describing the geochronological results and added this information in the main text (e.g., Lines 323-324).

The authors also cite Fig. S9 as containing additional evidence for abundant metamorphic-derived zircons in the Wufoshan Group (lines 286—287) but do not discuss the significance of the data presented in this supplementary figure. Fig S9 summarises REE patterns for zircons from the Wufoshan Group and shows that some grains have HREE-depleted signatures, which are typical of zircon that has co-crystallised with garnet, potentially indicating high-P metamorphism. However, these REE-patterns are not tied to the corresponding U-Pb age of the grains. Furthermore, the HREE-depleted measurements apparently come from the cores of zircon grains that are surrounded by metamorphic rims, which the authors appear link to the main Paleoproterozoic metamorphic event. Based on this data, I have two queries:

(1) How can the authors be certain these HREE-depleted metamorphic zircons aren't pre-2.0 Ga grains and hence actually provide evidence for high-P metamorphism related to the earlier (Neoproterozoic) orogenic event(s). I note from Fig. 6F that the Wufoshan Group contains numerous pre-2.0 Ga detrital grains. If the authors could demonstrate that all of the HREE-depleted zircons in the Wufoshan Group are ~1.7-2.0 Ga, it would strengthen the interpretation that the Paleoproterozoic metamorphic event was characterised by higher pressure metamorphism compared to earlier orogenic events.

Thank you for this insightful comment and suggestion. We added more CL images showing the structure of zircons with HREE-depleted characteristics (low Lu/Gy_N ratios, Fig. S8 below) and labelled the ages and Th/U and (Lu/Dy)_N ratios of these zircons (see new Fig. 6 below). Except

for one older metamorphic zircon with an age of 2.15 Ga, the ages of the rest zircons are concentrated at ca. 2.0–1.9 Ga. Circa 2.15 Ga metamorphism is reported by Trap et al. (2012) and Wang et al. (2023) from the central orogen of the NCC, and 2.0–1.8 Ga metamorphism is widespread in this orogen. As we discussed in the paper, the HREE-depleted signature suggests that these zircon grains are co-crystallized with garnet, and may have undergone HP metamorphism (Zhu et al., 2021, EPSL), indicating that these zircon grains in the Wufoshan Group recorded the Paleoproterozoic metamorphism.

Updated Supplementary Figure S8 shows the CL images of some typical zircons in the Wufoshan Group, and chondrite normalized REE patterns of zircon from the Wufoshan, Songshan and Dengfeng.

New Fig. 6 shows the relation between HREE-depleted metamorphic garnet signature (MGS) in zircon with age. Except for two Archean grains with MGS, others are early Proterozoic in age (dominantly 2.0–1.9 Ga).

(2) I am curious as to why similar detrital zircon REE data has not been presented from the Songshan Group. A comparison of time-constrained REE data from detrital zircons from both the Wufoshan and Songshan groups would make for a much more balanced comparison of the metamorphic style of Neoproterozoic vs. Paleoproterozoic orogenesis based on detrital zircon record (i.e., demonstrate that Neoproterozoic detrital zircon populations lack HREE-depleted signatures, whereas Paleoproterozoic zircon populations do contain them).

Thank you for the good suggestion. As shown above, we added all REE patterns from three units (Fig. S8) and added a figure in the main text (Fig. 6) to show the REE data [Lu and (Lu/Dy)_N that signify the metamorphic garnet signature, MGS] of detrital zircons from the Wufoshan Group, Songshan Group and Dengfeng Complex, along with REE data sets from the global detrital zircons as a background for comparison.

Zircons from the Songshan and Wufoshan groups have only three Neoproterozoic analyses with an HREE-depleted signature, and the HREE-depleted analyses are dominated by Paleoproterozoic ages (ca. 2.0–1.8 Ga), supporting our interpretation about the different styles of metamorphism (and orogenic styles). This is also consistent with the distribution of MGS in the global detrital zircon dataset (Fig. 6d).

Minor comments (Line-by-line)

Line 1 (title): I would remove ‘dramatic’ from the title. Much of the evidence for a change in orogenic style in the NCC is from models and datasets beyond the foreland basin deposits that this paper focuses on. I think the evidence from the sedimentary record is more subtle but compelling nonetheless (as is always the case for reconstructing orogenic histories from sedimentary rocks).

Thanks, good point. We revised the title and deleted the ‘dramatic’.

Line 53—54: Somewhat redundant to describe an ‘Alpine-style’ fold-and-thrust belt associated with a ‘Himalayan-style’ collisional orogeny. Describing it as a fold-and-thrust belt developed during a Himalayan- (or Alpine) style collision would suffice.

Thanks for this good point. We have changed it to ‘Himalayan-style’.

Line 70: Replace ‘loosely’ with ‘incompletely’ documented.

Done.

Line 113: The clasts (not ‘gravels’) in the conglomerate are aligned with the tectonic foliation.

Revised.

Lines 133: suggest rewording: The sandstones is composed of well-rounded detrital quartz grains that contain dusted rims and quartz overgrowths.

Thanks. Revised.

Line 216: Typo: 'provenances' should be provenance.

Revised.

Line 217: Typo: 'filtration' should be filtering.

Revised.

Line 276: '...could be linked to a secular change of orogenesis'. This clause isn't needed here. The point being made is that rutile forms in high-pressure metamorphic rocks. This is true regardless of when high-pressure rocks first appeared in the geological record (which is more relevant when this point is discussed later in the paper).

Agree, we deleted this.

Line 278: Typo: 'event' should be events.

Revised.

Line 282: Typo: 'metamorphic origins' should be metamorphic origin.

Revised.

Line 288: 'indicate' should be replaced with '....are interpreted to reflect....' or 'record'.

Good point. Revised.

Line 295: unclear what is meant by '....where the largest accretion and crustal growth occurred'. Largest accretion and growth compared to what? The local geology or the globally? Or do you

mean Altaid-style orogens are the sites of significant crustal accretion and growth?

Thanks, and yes, the Altai is the largest Phanerozoic accretionary orogen, where the juvenile crust was accreted. We revised this sentence to make it clearer.

Line 299: final sentence of this paragraph is vague. Do you mean rutile and low-Lu and low-Lu/Dy zircons are rare at this time in the global detrital record? If you want to make this point, you need to explain the significance of low-Lu and low-Lu/Dy zircons.

Thanks for this good comment. We revised the sentence and added a relevant reference to indicate that the low-Lu and low-Lu/Gd zircons can signify the garnet signature that formed generally >1.2 GPa (Zhu et al., 2022, EPSL). We also supplemented a figure (Fig. 6) to show this.

Zhu, Z., Campbell, I. H., Allen, C. M., Brocks, J. J. & Chen, B. The temporal distribution of Earth's supermountains and their potential link to the rise of atmospheric oxygen and biological evolution. *Earth Planet. Sci. Lett.* **580**, 117391 (2022).

Line 305: I'm not convinced that high-pressure granulites (and to some extent eclogites) provide evidence of 'cold subduction'. Granulites aren't common in 'cold' subduction zones and perhaps only Lawsonite eclogites provide compelling evidence for 'cold' subduction. If there are metamorphic rocks that do record low T/P thermal gradients in the NCC, these should be highlighted.

Thank you for this good suggestion. Xu et al. (2018, Nature Communications) reported low-temperature and high-pressure eclogite xenoliths (~560 °C/2.6 GPa, ca. 1.85 Ga) from carbonatite dyke, which has been regarded as an important piece of evidence for low T/P thermal gradients. We have added this reference.

Xu, C. et al. Cold deep subduction recorded by remnants of a Paleoproterozoic carbonated slab. *Nat. Commun.* **9**, 2790 (2018).

Line 317—319: As written it this sentence implies that only the tectonic events in the NCC influenced the wider Earth system. I think the authors mean that the changes recorded in the NCC are typical of those preserved in many early Paleoproterozoic terranes, suggesting a global

link between a transition in tectonic regime and surficial environments. As such, it would be helpful to cite some other examples where similar Archean-Paleoproterozoic tectonic transitions are preserved in the structural, sedimentary, magmatic record (e.g., Western Australia, North America, South Africa...).

Good point, North China records the cratonization at 2.5-2.45 Ga due to collision between arcs and micro-continent blocks. Following this, widespread clastic rocks were developed along foreland and cratonic margins. The record of sedimentation increased in most cratons worldwide, such as in India, North America, Western Australia, and South Africa (e.g., Cawood et al., 2018 and references therein). We revised this sentence and added relevant references.

Cawood, P. A. et al. Geological archive of the onset of plate tectonics. *Phil. Trans. R. Soc. A.* **376**, 20170405 (2018).

Line 319—320: Are you saying this style of late Archean cratonisation is typical of the NCC or all cratons? If the latter, need to support with references.

Thank you for this good suggestion. The Neoproterozoic cratonization should have been widespread in most cratons, as there were many Neoproterozoic to early Paleoproterozoic clastic sedimentary rocks overlain the Archean basement rocks. We added relevant references from other cratons here.

Line 325: Typo: ‘Nutrition’ should be nutrients. For those not familiar with biochemical cycles, it would be useful to elaborate on what these nutrients actually are and how they form an important component of orogenic rocks.

Thank you. We corrected the typo, and added these key elements (e.g., P, Ca, Mg), which are the key nutrients thought to limit primary productivity on geological timescales.

Line 334—336: Might be helpful to remind the readers that these are quartz-rich clastic rocks, which provides evidence that the more labile minerals in their source TTGs/greenstones were indeed destroyed during weathering.

Thank you for this good suggestion. We added this point.

Line 354—356: A citation to Cawood et al. 2022 *Reviews of Geophysics*, v. 60 (4) would be appropriate here.

Agree. We also cited this nice review in Introduction section.

Line 358: ‘higher peak metamorphic pressures from minerals or increased high-pressure rock record’. This is vague, aren’t these the same thing? i.e., high-pressure rocks are defined by having high-pressure minerals.

Thanks for this good comment. We revised this sentence accordingly.

Line 362—367: Unclear if this is a general statement about all orogens or just the study area.

We revised this sentence. This statement about the orogenic characteristics can be applicable to the majority of global Paleoproterozoic orogens, as more and more high-pressure rocks have been recognized (e.g., Weller and St-Onge, 2017, NG; Brown and Johnson, 2018 for a summary). Some key features such as the foreland compressional deformation documented in this study are highlighted.

Weller, O. M. & St-Onge, M. R. Record of modern-style plate tectonics in the Palaeoproterozoic Trans-Hudson orogen. *Nat. Geosci.* **10**, 305-311 (2017).

Brown, M. & Johnson, T. Secular change in metamorphism and the onset of global plate tectonics. *Am. Miner.* **103**, 181-196 (2018).

Line 368—377: The concluding paragraph linking the stabilisation of continents to the boring billion and the appearances of eukaryotes is essentially a throw-away line. In the context of this paper, the link between continental stabilisation, the boring billion, and eukaryote evolution is not clear. Is it to do with ocean chemistry? tectonics? What new insights on these topics does this paper make? A stronger finish would be to reinforce the sedimentary response to changing orogenic styles, which is really what this paper does a nice job of doing.

Thank you for the good comment and suggestion. The original intention in the last paragraph was to emphasize that the collapse of the Paleoproterozoic Himalayan-style collisional orogens led to continental stabilisation and sedimentation such as those in the Wufoshan Group, which

deposited along orogenic and cratonic margin or interior. We shortened this paragraph and added some references.

We add a concluding paragraph to highlight the sedimentary response to the orogenic styles as below:

“In conclusion, this study provides new constraints on the depositional-to-deformational evolution of the Paleoproterozoic Songshan foreland — one of the oldest orogenic forelands in the North China Craton, from sedimentary and deformational perspectives. Importantly, it underscores the significance of multi-mineral geochronological analysis of key minerals from sedimentary archives in ancient orogenic forelands, which provide clues as to how sedimentation and metamorphism responded to changing orogenic style. Our research has shed new light on changes in tectonic processes during the Archean-Proterozoic transition and their influence on global biogeochemical cycles, providing an innovative method for investigating the evolution of foreland sedimentary basins. Additionally, our findings offer fresh perspectives on the dynamic interplay between solid and surficial Earth processes during the Paleoproterozoic, contributing to a deeper understanding of our planet’s evolution towards a more habitable environment.
”

Line 379: (Methods). No data for reference materials is presented in the supplementary material. It is stated that ‘all ages are consistent with recommended values’ but the original data for the reference materials must be included so their veracity can be assessed independently.

Thank you for this important comment, we added the standard information and references accordingly.

General comments on grammar

- The definitive article (the) is used incorrectly in several places throughout the manuscript. I encourage the native English-speaking co-authors or journal type editors to revise.

Thank you. we carefully checked the grammar issue.

- A more pedantic point, but the use of forward slashes (/) obscures the intended meaning of the

text in several places. It's not clear if the forward slash means 'or', 'and', or is taking the place of a hyphen. For example, arc/micro-continent collision. Is this describing a collision between an arc and a microcontinent (arc-microcontinent collision), or is it indicating there were multiple collisions, some involving arcs and microcontinents, others involving only arcs or only microcontinents? I suggest replacing '/' with 'and' or with 'or' for clarity.

Good point. We have replaced the forward slashes with the 'and' 'or' or 'hyphen' in most places.

We are deeply appreciative of the comprehensive and constructive comments and suggestions provided by Reviewer #3. These inputs have played a pivotal role in enhancing the quality of our manuscript.

REVIEWERS' COMMENTS

Reviewer #1 (Remarks to the Author):

I have carefully read through the detailed responses to reviewers and the revised manuscript provided by the authors. I think they have done a thorough job in responding to the reviewers' comments, with an excellent level of detail. It is my view that the reviewers' questions have been appropriately answered, and their comments and suggestions, addressed and incorporated. I think the manuscript is now ready for publication in NC. I would like to thank the authors for the thoughtful and detailed way in which they responded to the reviewers' concerns.

Reviewer #2 (Remarks to the Author):

1. I don't think the authors seriously responded to the comments and didn't revised the paper to address the concerns; the authors tried to avoid the main issues.

2. Again, one of the main problems is that this work in this paper is not novel, similar concepts and have been discussed in the previous publications. Therefore, I do not think the paper meets the standard and level of the journal of Nature Communication. The paper is more suitable for publication in journal of Precambrian Research, Tectonics and et al.

3. Too many papers showed the regional evolution models of the North China Craton during Neoproterozoic and Paleoproterozoic similar to this paper, whether it's once or twice collision and when. There's no need to list them all here. This study contributes nothing new among these models and don't clarify these controversies.

4. Typos and weird expression here and there.

I strongly reject the publication of this revised manuscript in Nature Communications.

Reviewer #3 (Remarks to the Author):

The authors have done a thorough job of revising this manuscript. All of my comments on the original manuscript have been clearly and comprehensively addressed.

It is particularly satisfying to see that the authors have presented new observations and analysis in addressing my comments, which further strengthen what was already an excellent study.

This is an interesting and engaging piece of work that will make a valuable contribution to Nature Communications. I recommend the revised manuscript be accepted.

RESPONSE TO REVIEWERS' COMMENTS (REPLY IN BLUE)

Reviewer #1 (Remarks to the Author):

I have carefully read through the detailed responses to reviewers and the revised manuscript provided by the authors. I think they have done a thorough job in responding to the reviewers' comments, with an excellent level of detail. It is my view that the reviewers' questions have been appropriately answered, and their comments and suggestions, addressed and incorporated. I think the manuscript is now ready for publication in NC. I would like to thank the authors for the thoughtful and detailed way in which they responded to the reviewers' concerns.

We deeply appreciate the Reviewer #1 for the constructive comments and positive recommendation during the past two rounds of reviews.

Reviewer #2 (Remarks to the Author):

. I don't think the authors seriously responded to the comments and didn't revised the paper to address the concerns; the authors tried to avoid the main issues.

We have carefully revised the manuscript and addressed the concerns from all reviewers in the first revision, as summarized by other reviewers. In particular, we have addressed all concerns and issues raised by this reviewer in the main text and response letter in the first revision, with emphasizing the aim and key innovation points of this manuscript. No issue has been avoided.

2. Again, one of the main problems is that this work in this paper is not novel, similar concepts and have been discussed in the previous publications. Therefore, I do not think the paper meets the standard and level of the journal of Nature Communication. The paper is more suitable for publication in journal of Precambrian Research, Tectonics and et al.

In this study, we use the integrated field, structural and multi-mineral geochronological data sets from the one of oldest foreland successions in the North China Craton to better constrain the distinctive styles of two stages of orogenesis. The detrital zircon U-Pb data constrain the maximum depositional age of the succession. The structural data reveal the Alpine style fold-and-thrust structures. The combined usage of detrital rutile and zircon from different sedimentary successions (Songshan vs. Wufoshan groups) provides new clues and insights into the metamorphic patterns of the source regions, thus reflecting the different orogenic styles. We further link the change of orogenesis with silicate weathering and near-surface environment change.

The reviewer seems ignore the new perspectives (e.g., sedimentary and structural) and methods (e.g., multi-mineral geochronology, big-data) on the general scientific questions (e.g., the change of orogenic and plate tectonic styles, interplay between solid and surface systems) presented in this study.

3. Too many papers showed the regional evolution models of the North China Craton during Neoproterozoic and Paleoproterozoic similar to this paper, whether it's once or twice collision and when. There's no need to list them all here. This study contributes nothing new among these models and don't clarify these

controversies.

As we replied before, there are two episodes of tectono-thermal events documented in the North China Craton, which have been linked with growth and assembly of the craton, providing a unique natural laboratory to explore the styles of orogenesis and plate tectonics.

Except for the general implications, our results contribute to at least three aspects on the evolution of the North China Craton: **1)** constraints on the age span of deposition of the Songshan Group, providing further sedimentary evidence for the Altiid-style terrane assembly and arc-microcontinent collision in the end-Archean. **2)** determination of Paleoproterozoic metamorphism and Alpine-style deformation in the Dengfeng-Songshan segment, indicating this region underwent overprinting during the Paleoproterozoic Himalayan-style orogenesis. **3)** multi-mineral geochronology and detrital rutile and zircon reveal different metamorphic patterns in the Neoproterozoic and Paleoproterozoic, consistent with distinctive orogenic styles.

These new findings suggest that the tectonic evolution of the North China Craton is very complex, and available popular tectonic models may need further improvement, which, however, is not the focus of this manuscript where we concentrate on the orogenic styles.

4. Typos and weird expression here and there.

This is an ambiguous comment. The text has been carefully checked by all coauthors (including three native English speakers) in the previous version. Before submission of this new version, we also carefully checked the language again.

I strongly reject the publication of this revised manuscript in Nature Communications.

We thank Reviewer #2 for his/her critical comments during past two rounds of review, which have been useful for us to clarify and improve the work.

Reviewer #3 (Remarks to the Author):

The authors have done a thorough job of revising this manuscript. All of my comments on the original manuscript have been clearly and comprehensively addressed.

It is particularly satisfying to see that the authors have presented new observations and analysis in addressing my comments, which further strengthen what was already an excellent study.

This is an interesting and engaging piece of work that will make a valuable contribution to Nature Communications. I recommend the revised manuscript be accepted.

We deeply appreciate Reviewer #3 for the constructive comments and positive recommendations during the past two rounds of reviews. The comments have been very helpful for us to clarify and improve the quality of the manuscript.